# How to measure deep uncertainty estimation performance and which models are naturally better at providing it

## Abstract

When deployed for risk-sensitive tasks, deep neural networks (DNNs) must be equipped with an uncertainty estimation mechanism. This paper studies the relationship between deep architectures and their training regimes with their corresponding uncertainty estimation performance. We consider both in-distribution uncertainties ("aleatoric" or "epistemic") and class-out-of-distribution ones. Moreover, we consider some of the most popular estimation performance metrics previously proposed including AUROC, ECE, AURC, and coverage for selective accuracy constraint. We present a novel and comprehensive study carried out by evaluating the uncertainty performance of 484 deep ImageNet classification models. We identify numerous and previously unknown factors that affect uncertainty estimation and examine the relationships between the different metrics. We find that distillation-based training regimes consistently yield better uncertainty estimations than other training schemes such as vanilla training, pretraining on a larger dataset and adversarial training. We also provide strong empirical evidence showing that ViT is by far the most superior architecture in terms of uncertainty estimation performance, judging by any aspect, in both in-distribution and class-out-of-distribution scenarios. We learn various interesting facts along the way. Contrary to previous work, ECE does not necessarily worsen with an increase in the number of network parameters. Likewise, we discovered an unprecedented 99% top-1 selective accuracy at 47% coverage (and 95% top-1 accuracy at 80%) for a ViT model, whereas a competing EfficientNet-V2-XL cannot obtain these accuracy constraints at any level of coverage.

## 1 Introduction

Deep neural networks (DNNs) show great performance in a wide variety of application domains including computer vision, natural language understanding and audio processing. Successful deployment of these models, however, is critically dependent on providing an effective *uncertainty estimation* of their predictions in the form of some kind of *selective prediction* or providing a probabilistic confidence score for their predictions.

But how should we evaluate the performance of uncertainty estimation? Let us consider two classification models for the stock market that predict whether a stock's value is about to increase, decrease or remain neutral (three-class classification). Suppose that model A has a 95% true accuracy, and generates a confidence score of 0.95 on every prediction (even on misclassified instances); model B has a 40% true accuracy, but always gives a confidence score of 0.6 on correct predictions, and 0.4 on incorrect ones. Model B can be utilized easily to generate perfect investment decisions. Using *selective prediction* (Geifman & El-Yaniv, 2017), Model B will reject all investments on stocks whenever the confidence score is 0.4. While model A offers many more investment opportunities, each of its predictions carries a 5% risk of failure.

Among the various metrics proposed for evaluating the performance of uncertainty estimation are: *Area Under the Receiver Operating Characteristic* (AUROC or AUC), *Area Under the Risk-Coverage curve* (AURC) (Geifman et al., 2018), selective risk or coverage for a *selective accuracy constraint* (SAC), *Negative Log-likelihood* (NLL), *Expected Calibration Error* (ECE), which is of-

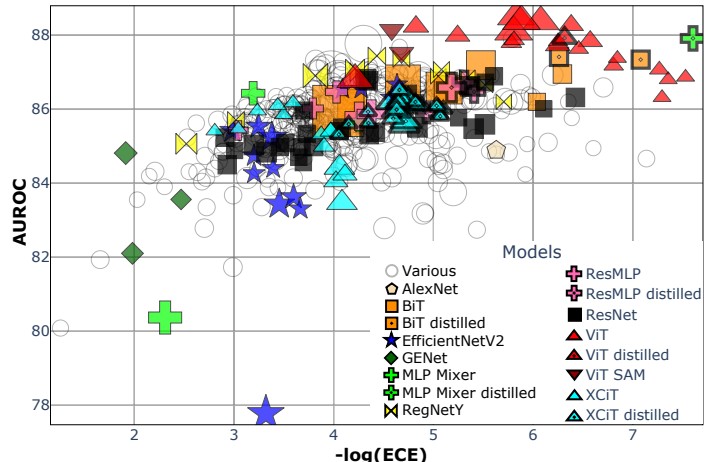

Figure 1: A comparison of 484 models by their AUROC (×100, higher is better) and -log(ECE) (higher is better) on ImageNet. Each marker's size is determined by the model's number of parameters. A full version graph is given in Figure 8. Distilled models are better than non-distilled ones. ViT models are naturally better at all aspects of uncertainty estimation, while EfficientNet-V2 and GENet models are worse.

ten used for evaluating a model's *calibration* (see Section 2) and *Brier score* (Brier, 1950). All these metrics are well known and are often used for comparing the uncertainty estimation performance of models (Moon et al., 2020; Nado et al., 2021; Maddox et al., 2019; Lakshminarayanan et al., 2017). Somewhat surprisingly, NLL, Brier, AURC, and ECE all fail to reveal the uncertainty superiority of Model B in our investment example (see Appendix A for the calculations). Both AUROC and SAC, on the other hand, reveal the advantage of Model B perfectly (see Appendix A for details). It is not hard to construct counter examples where these two metrics fails and others (e.g., ECE) succeed. The risk-coverage (RC) curve (El-Yaniv & Wiener, 2010) is perhaps one of the most informative and practical representations of the overall uncertainty profile of a given model.

In general, though, two RC curves are not necessarily comparable if one does not fully dominate the other (see Figure 2). The advantage of scalar metrics such as the above is that they summarize the model's overall uncertainty estimation behavior by reducing it to a single scalar. When not carefully chosen, however, these reductions could result in a loss of vital information about the problem (for example, reducing an RC curve to an AURC does not show that Model B has an optimal 0 risk if the coverage is smaller than 0.4). Thus, the choice of the "correct" single scalar performance metric unfortunately must be task-specific. When comparing the uncertainty estimation performance of deep architectures that exhibit different accuracies, we find that AUROC and SAC can effectively "normalize" accuracy differences that plague the usefulness of other metrics (see Section 2). This normalization is essential to our study where we compare uncertainty performance of hundreds of models that can greatly differ in their accuracies.

In applications where risk (or coverage) constraints are dictated (Geifman & El-Yaniv, 2017), the most straightforward and natural metric is the SAC (or selective risk), which directly measures the coverage (resp., risk) given at the required level of risk (resp., coverage) constraint. We demonstrate this in Appendix J, evaluating which models give the most coverage for a SAC of 99%. Sometimes, however, such constraints are unknown in advance, or even irrelevant, e.g., the constructed model should serve a variety of risk constraint use cases, or the model may not be allowed to abstain from predicting at all.

In this paper we conduct a comprehensive study of DNNs' ability to estimate uncertainty by evaluating 484 models pretrained on ImageNet (Deng et al., 2009), taken from the PyTorch and timm respositories (Paszke et al., 2019; Wightman, 2019). We identify the main factors contributing to or harming the confidence ranking of predictions ("ranking" for short), calibration and selective prediction. Furthermore, we also consider the source of uncertainty as either *internal* (stemming from either the *aleatoric* or *epistemic* uncertainty of the model (Kiureghian & Ditlevsen, 2009)) or *external* (originating from unseen or unknown class-out-of-distribution (C-OOD) data) and evaluate

these models in multiple ways. After first evaluating models solely on in-distribution (ID) data, we then define and test two ways of evaluating C-OOD data, each of which also divides the data into different groups by how difficult it is for the model to distinguish instances as external.

Our study lead to quite a few new observations and conclusions; (1) Training regimes incorporating any kind of *knowledge distillation* (KD) (Hinton et al., 2015) leads to DNNs with improved uncertainty estimation performance evaluated by any metric, in both internal and external settings (i.e., leading also to better C-OOD detection), more than by using any other training tricks (such as pre-training on a larger dataset, adversarial training, etc.). (2) Some architectures are naturally superb at all aspects of uncertainty estimation and in all settings, e.g., vision transformers (ViTs) (Dosovitskiy et al., 2020; Steiner et al., 2021), while other architectures tend to perform worse, e.g., EfficientNet-V2 and GENet (Tan & Le, 2021; Lin et al., 2020). These results are visualized in Figure 1. (3) The superiority of ViTs remains even when the comparison considers the models' sizes—meaning that for any size, ViTs outperform the competition in uncertainty estimation performance, as visualized in Appendix B in Figures 9 and 10. (4) The simple post-training calibration method of *temperature scaling* (Guo et al., 2017), which is known to improve ECE, for the most part also improves ranking (AUROC) and selective prediction—meaning not only does it calibrate the probabilistic estimation for each individual instance, but it also improves the partial order of all instances induced by those improved estimations, pushing instances more likely to be correct to have higher confidence than instances less likely to be correct (see Section 3). (5) Contrary to previous work by Guo et al. (2017), we observe that while there is a strong correlation between accuracy/number of parameters and ECE or AUROC within each specific family of models of the same architecture, the correlation flips between a strong negative and a strong positive correlation depending on the type of architecture being observed. For example, as ViT architectures increase in size and accuracy, their ECE deteriorates while their AUROC improves. The exact opposite, however, could be observed in XCiTs (El-Nouby et al., 2021) as their ECE improves with size while their AUROC deteriorates. (see Appendix G). (6) The best model in terms of AUROC or SAC is not always the best in terms of calibration, as illustrated in Figure 1, and the trade-off should be considered when choosing a model based on its application. Due to lack of space, a number of additional interesting observations are briefly mentioned in the paper without supporting empirical evidence (which is provided in the appendix).

## 2 HOW TO EVALUATE DEEP UNCERTAINTY ESTIMATION PERFORMANCE

Let $\mathcal{X}$ be the input space and $\mathcal{Y}$ be the label space. Let $P(\mathcal{X}, \mathcal{Y})$ be an unknown distribution over $\mathcal{X} \times \mathcal{Y}$. A model $f$ is a prediction function $f : \mathcal{X} \to \mathcal{Y}$, and its predicted label for an image $x$ is denoted by $\hat{y}_f(x)$. The model's *true* risk w.r.t. $P$ is $R(f|P) = E_{P(\mathcal{X}, \mathcal{Y})}[\ell(f(x), y)]$, where $\ell : \mathcal{Y} \times \mathcal{Y} \to \mathbb{R}^+$ is a given loss function, for example, 0/1 loss for classification. Given a labeled set $S_m = \{(x_i, y_i)\}_{i=1}^m \subseteq (\mathcal{X} \times \mathcal{Y})$, sampled i.i.d. from $P(\mathcal{X}, \mathcal{Y})$, the *empirical risk* of model $f$ is $\hat{r}(f|S_m) \triangleq \frac{1}{m} \sum_{i=1}^m \ell(f(x_i), y_i)$. Following Geifman et al. (2018), for a given model $f$ we define a *confidence score* function $\kappa(x, \hat{y}|f)$, where $x \in \mathcal{X}$, and $\hat{y} \in \mathcal{Y}$ is the model's prediction for $x$, as follows. The function $\kappa$ should quantify confidence in the prediction of $\hat{y}$ for the input $x$, based on signals from model $f$. This function should induce a partial order over instances in $\mathcal{X}$, and is not required to distinguish between points with the same score.

The most common and well-known $\kappa$ function for a classification model $f$ (with softmax at its last layer) is its softmax response values: $\kappa(x, \hat{y}|f) \triangleq f(x)_{\hat{y}}$ (Cordella et al., 1995; De Stefano et al., 2000). While this is the main $\kappa$ we evaluate, we also test the popular uncertainty estimation technique of *Monte-Carlo dropout* (MC-Dropout) (Gal & Ghahramani, 2016), which is motivated by Bayesian reasoning. Although these methods use the direct output from $f$, $\kappa$ could be a different model unrelated to $f$ and unable to affect $f$'s predictions. Note that to enable a probabilistic interpretation, $\kappa$ can only be calibrated if its values reside in $[0, 1]$ whereas for ranking and selective prediction any value in $\mathbb{R}$ can be used.

A *selective model* $f$ (El-Yaniv & Wiener, 2010; Chow, 1957) uses a *selection function* $g : \mathcal{X} \to \{0, 1\}$ to serve as a binary selector for $f$, enabling it to abstain from giving predictions for certain inputs. $g$ can be defined by a threshold $\theta$ on the values of a $\kappa$ function such that $g_\theta(x|\kappa, f) = \mathbb{1}[\kappa(x, \hat{y}_f(x)|f) > \theta]$. The performance of a selective model is measured using coverage and risk, where *coverage*, defined as $\phi(f, g) = E_P[g(x)]$, is the probability mass of the non-rejected instances in $\mathcal{X}$. The *selective risk* of the selective model $(f, g)$ is defined as $R(f, g) \triangleq \frac{E_P[\ell(f(x), y)g(x)]}{\phi(f, g)}$. These

quantities can be evaluated empirically over a finite labeled set $S_m$, with the *empirical coverage* defined as $\hat{\phi}(f, g|S_m) = \frac{1}{m}\sum_{i=1}^{m} g(x_i)$, and the *empirical selective risk* defined as $\hat{r}(f, g|S_m) \triangleq \frac{\frac{1}{m}\sum_{i=1}^{m} \ell(f(x_i), y_i)g(x_i)}{\hat{\phi}(f, g|S_m)}$. Similarly, SAC is defined as the largest coverage available for a specific accuracy constraint. A way to visually inspect the behavior of a $\kappa$ function for selective prediction can be done using an RC curve—a curve showing the selective risk as a function of coverage, measured on some chosen test set; see Figure 2 for an example.

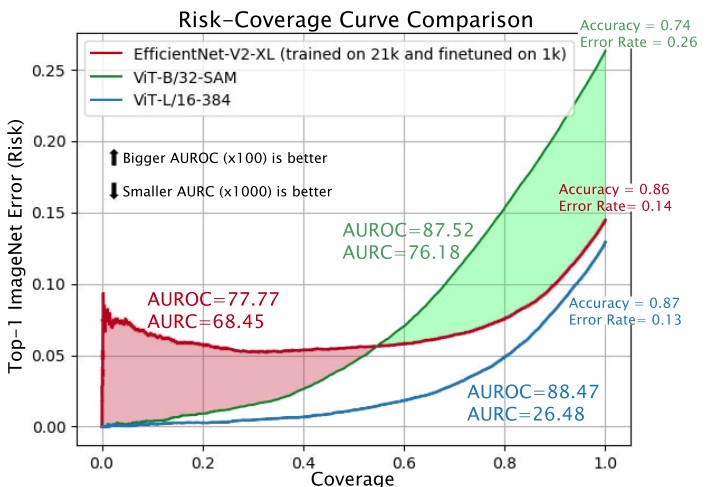

Figure 2: A comparison of RC-curves made by the best (ViT-L/16-384) and worst (EfficientNet-V2-XL) models we evaluated in terms of AUROC. Comparing ViT-B/32-SAM to EfficientNet-V2 exemplifies the fact that neither accuracy nor AURC reflect selective performance well enough.

The AURC and E-AURC metrics were defined by Geifman et al. (2018) for quantifying the selective quality of $\kappa$ functions via a single number, with AURC being defined as the area under the RC curve. AURC, however, is very sensitive to the model's accuracy, and in an attempt to mitigate this, E-AURC was suggested. The latter also suffers from sensitivity to accuracy, as we demonstrate in Appendix C. Let us consider the two models in Figure 2 for risk-sensitive deployment; EfficientNet-V2-XL (Tan & Le, 2021) and ViT-B/32-SAM (Chen et al., 2021a). While the former model has better overall accuracy and AURC (metrics that could lead us to believe the model is best for our needs), it cannot guarantee a Top-1 ImageNet selective accuracy above 95% for any coverage. ViT-B/32-SAM, on the other hand, can provide accuracies above 95% for all coverages below 50%.

When there are requirements for specific coverages, the most direct metric to utilize would be the matching selective risks, by which we can select the model offering the best performance for our task. If instead a specific range of coverages is specified, we could measure the area under the RC curve for those coverages: $\text{AURC}_{\mathcal{C}}(\kappa, f|S_m) = \frac{1}{|\mathcal{C}|}\sum_{c\in\mathcal{C}} \hat{r}(f, g_c|S_m)$, with $\mathcal{C}$ being those required coverages. Lastly, if a certain accuracy constraint is specified, the chosen model should be the one providing the largest coverage for that constraint (the largest coverage for a certain SAC).

Often, these requirements are not known or can change as a result of changing circumstances or individual needs. Also, using metrics sensitive to accuracy such as AURC makes designing architectures and methods to improve $\kappa$ very hard, since an improvement in these metrics could be attributed to either an increase in overall accuracy (if such occurred) or to a real improvement in the model's "metacognition". Lastly, some tasks might not allow the model to abstain from making predictions at all, but instead require interpretable and well-calibrated probabilities of correctness, which could be measured using ECE.

## 2.1 MEASURING RANKING AND CALIBRATION

A $\kappa$ function is not necessarily able to change the model's predictions. Thus, its means for improving the selective risk is by ranking correct and incorrect predictions better, inducing a more accurate partial order over instances in $\mathcal{X}$. Thus, for every two random samples $(x_1, y_1), (x_2, y_2) \sim P(\mathcal{X}, \mathcal{Y})$

and given that $\ell(f(x_1), y_1) > \ell(f(x_2), y_2)$, the *ranking* performance of $\kappa$ is defined as the probability that $\kappa$ ranks $x_2$ higher than $x_1$:

$$\mathbf{Pr}[\kappa(x_1, \hat{y}|f) < \kappa(x_2, \hat{y}|f)|\ell(f(x_1), y_1) > \ell(f(x_2), y_2)] \tag{1}$$

We discuss this definition in greater detail in Appendix D. The AUROC metric is often used in the field of machine learning. When the 0/1 loss is in play, it is known that AUROC in fact equals the probability in Equation (1) (Fawcett, 2006) and thus is a proper metric to measure ranking in classification (AKA discrimination). AUROC is furthermore equivalent to the Goodman and Kruskal's $\gamma$-*correlation* Goodman & Kruskal (1954), which for decades has been extensively used to measure ranking (known as "resolution") in the field of metacognition Nelson (1984). The precise relationship between $\gamma$-correlation and AUROC is $\gamma = 2 \cdot \text{AUROC} - 1$ (Higham & Higham, 2018). We note also that both the $\gamma$-correlation and AUROC are nearly identical or closely related to various other correlations and metrics; $\gamma$-correlation (AUROC) becomes identical to Kendall's $\tau$ (up to a linear transformation) in the absence of tied values. both metrics are also closely related to *rank-biserial correlation*, the *Gini coefficient* (not to be confused with the measure from economics) and the *Mann–Whitney U test*, hinting at their importance and usefulness in a variety of fields and settings. In Appendix E, we briefly compare the ranking performance of neural networks and humans based on metacognitive research and address a criticism of using AUROC to measure ranking in Appendix F

The most widely used metric for calibration is ECE (Naeini et al., 2015). For a finite test set of size $N$, ECE is calculated by grouping all instances into $m$ interval bins (such that $m \ll N$), each of size $\frac{1}{m}$ (the confidence interval of bin $B_j$ is $(\frac{j-1}{m}, \frac{j}{m}]$). With $\text{acc}(B_j)$ being the mean accuracy in bin $B_j$ and $\text{conf}(B_j)$ being its mean confidence, ECE is defined as

$$ECE = \sum_{j=1}^{m} \frac{|B_j|}{N} \sum_{i \in B_j} \left| \frac{\mathbb{1}[\hat{y}_f(x_i) = y_i]}{|B_j|} - \frac{\kappa(x, \hat{y}_f(x_i)|f)}{|B_j|} \right| = \sum_{j=1}^{m} \frac{|B_j|}{N} \sum_{i \in B_j} |\text{acc}(B_j) - \text{conf}(B_j)|$$

Since ECE is widely accepted we use it here to evaluate calibration, and follow Guo et al. (2017) in setting the number of bins to $m = 15$. Many alternatives to ECE exist, to allow an adaptive binning scheme or to evaluate the calibration on the non-chosen labels as well (Nixon et al., 2019; Vaicenavicius et al., 2019). Relevant to our objective is that by using binning, this metric is not affected by the overall accuracy as is the Brier score, for example.

## 3  IN-DISTRIBUTION ANALYSIS

While AUROC and ECE are (negatively) correlated (they have a Spearman correlation of -0.5, meaning that generally as AUROC improves so does ECE), their agreement on the best performing model depends greatly on the architectural family in question. For example, the Spearman correlation between the two metrics evaluated on 28 undistilled XCiTs is 0.76 (meaning ECE deteriorates as AUROC improves), while for the 33 ResNets (He et al., 2015) evaluated, the correlation is -0.74. Another general observation is that, contrary to previous work by Guo et al. (2017) concerning ECE, the correlations between AUROC or ECE and the accuracy or the number of model parameters are nearly zero, although each family tends to have a strong correlation, either negative or positive. We include a family-based comparison in Appendix G for correlations between AUROC/ECE and accuracy, number of parameters and input size. These results suggest that while some architectures might utilize extra resources to achieve improved uncertainty estimation capabilities, other architectures do not and are even harmed in this respect.

We evaluated several training regimes: (1) Training that involves knowledge distillation in any form, including transformer-specific distillation (Touvron et al., 2020), knapsack pruning with distillation (in which the teacher is the original unpruned model) (Aflalo et al., 2020) and a pretraining technique which employs distillation (Ridnik et al., 2021); (2) adversarial training (Xie et al., 2019a; Tramèr et al., 2018); (3) pretraining on ImageNet21k ("pure", with no additions) (Tan & Le, 2021; Touvron et al., 2021a); and (4) various forms of weakly or semi-supervised learning (Mahajan et al., 2018b; Yalniz et al., 2019; Xie et al., 2019b). Of these methods, training methods incorporating distillation improve AUROC and ECE the most (see Figures 3 and 4). Moreover, distillation seems to greatly improve both metrics even when the teacher itself is much worse at both metrics. We discuss these effects in greater detail in Appendix H.

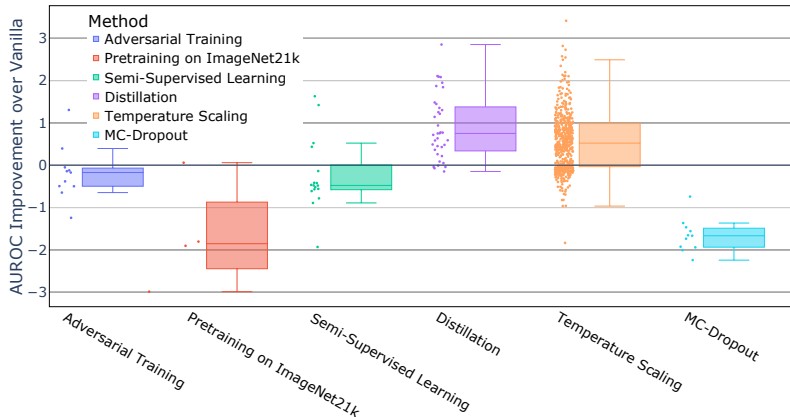

Figure 3: A comparison of different methods and their AUROC improvement relative to the same model's performance without employing the method. Markers above the x axis represent models that benefited from the evaluated method, and vice versa.

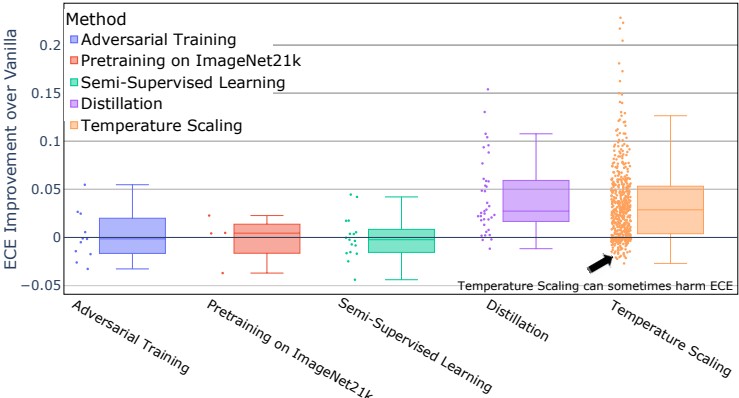

Figure 4: A comparison of different methods and their ECE improvement relative to the same model's performance without employing the method. Markers above the x axis represent models that benefited from the evaluated method, and vice versa. Temperature scaling can sometimes harm ECE, although its purpose is to improve it.

Evaluations of the simple post-training calibration method of temperature scaling (TS) (Guo et al., 2017), which is widely known to improve ECE without changing the model's accuracy, also revealed several interesting facts: (1) TS consistently and greatly improves AUROC and selective performance (see Figure 3)—meaning not only does TS calibrate the probabilistic estimation for each individual instance, but it also improves the partial order of all instances induced by those improved estimations. While TS is well known and used for calibration, to the best of our knowledge, its benefits for selective prediction were previously *unknown*. (2) While TS is usually beneficial, it could harm some models (see Figures 3 and 4). While it is surprising that TS–a calibration method– would harm ECE, this phenomenon is explained by the fact that TS optimizes NLL and not ECE (to avoid trivial solutions), and the two may sometimes misalign. (3) Models that benefit from TS in terms of AUROC tend to have been assigned a temperature smaller than 1 by the calibration process. This, however, does not hold true for ECE (see Figures 16 and 17 in Appendix I). (4) While all models usually improve with TS, the overall ranking of uncertainty performance between families tends to stay similar, with the worse (in terms of ECE and AUROC) models closing most of the gap between them and the mediocre ones.

The ViT architecture far surpasses any other family of models in terms of AUROC and ECE (see Figure 1; Figure 15 in Appendix I shows this is true even after using TS) as well as for the SAC of 99% we explored (see Figure 5 and Appendix J). Moreover, for any size, ViT models outperform their competition in all of these metrics (see Figures 9 and 10 in Appendix B and Figure 18 in

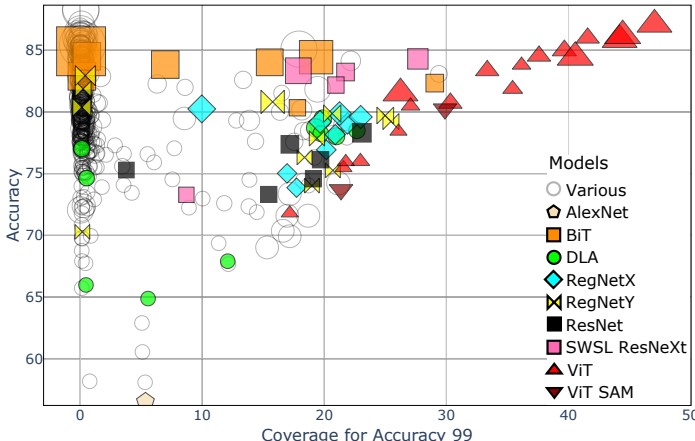

Figure 5: Comparison of models by their overall accuracy and the coverage they are able to provide a selective accuracy constraint of Top-1 99% on ImageNet. A higher coverage is better. Only ViT models are able to provide coverage beyond 30% for this constraint. They provide more coverage than any other model compared to their accuracy or size.

Appendix J). While most ViT models we evaluated were pretrained on ImageNet-21k and enjoyed augmentations tailored to them (Steiner et al., 2021), comparing them with the non-pretrained and not strongly augmented ViT SAMs (Chen et al., 2021a), which are consequently much worse in terms of accuracy than regular ViTs, confirms that the high performance in AUROC and SAC is not due to pretraining or augmentation. It does suggest that it is the architecture itself that is naturally superb at uncertainty estimation.

We also evaluate the AUROC performance of MC-Dropout using predictive entropy as its confidence score and 30 dropout-enabled forward passes. We do not measure its affects on ECE since entropy scores do not reside in $[0, 1]$. Using MC-Dropout causes a consistent drop in both AUROC and selective performance compared with using the same models with softmax as the $\kappa$ (see Appendix K and Figure 3). MC-Dropout's underperformance was also previously observed in (Geifman & El-Yaniv, 2017).

## 4 UNCERTAINTY DUE TO CLASS-OUT-OF-DISTRIBUTION

When the underlying distribution $P(x, y)$ used to train a model changes, we may no longer expect that the model will perform correctly. Changes in $P$ can be the result of many natural or adversarial processes such as natural deviation in the input space $\mathcal{X}$, noisy sensor reading of inputs, abrupt changes due to random events, newly arrived or refined input classes, etc. We distinguish between input distributional changes in $P_{X|Y}$ and changes in the label distribution. We focus on the latter case and consider the *class-out-of-distribution* (C-OOD) scenario where the label support set $\mathcal{Y}$ changes to a different set, $\mathcal{Y}_{OOD}$, which contains new classes that were not observed in training. We note that both aspects of distributional deviations have been considered in the literature (Hendrycks & Dietterich, 2019; Liang et al., 2017; Hendrycks et al., 2021), and a number of methods have been introduced to deal with these cases (Liang et al., 2017; Lee et al., 2018; Golan & El-Yaniv, 2018).

We consider the following *detection* task, in which our model is required to distinguish between samples belonging to classes it has seen in training, where $x \sim P(x|y \in \mathcal{Y})$, and novel classes, i.e., $x \sim P(x|y \in \mathcal{Y}_{OOD})$. We examine the detection performance of DNN classification models that use their confidence rate function $\kappa$ to detect OOD labels where the basic premise is that instances whose labels are in $\mathcal{Y}_{OOD}$ correspond to low $\kappa$ values.

A crucial question in any study of distributional deviations is what we choose as our experimental data to proxy meaningful deviations. For our study of C-OOD we introduce a novel method for generating C-OOD data with a controllable degree of *severity*. Let $\mathcal{Y}_{OOD}$ be a large set of OOD classes (e.g., labels from ImageNet-21k), and let $s(y|f, \kappa)$ be a *severity score* that reflects the difficulty of model $f$, which uses $\kappa$ to detect instances from class $y \in \mathcal{Y}_{OOD}$. Having defined a function

$s(y|f, \kappa)$ (see details below) we can build multiple C-OOD datasets with progressively increasing severity levels. Importantly, the resulting C-OOD data is specifically tailored to model $f$ itself (and its $\kappa$).

Given a model $f$ (and its $\kappa$), we define $s(y|f, \kappa)$ to be the average confidence given by $\kappa$ to samples from class $y \in \mathcal{Y}_{OOD}$. When considering ID instances we expect $\kappa$ to give high values for highly confident predictions. Therefore, the larger $s(y|f, \kappa)$ is, the harder it is for $\kappa$ to detect the OOD class $y$ among ID classes. We estimate $s(y|f, \kappa)$ for each class in ImageNet-21K (not in ImageNet-1K) using a sample from the class and take as C-OOD data a different sample from that class. Using $s$ we sub-sample 11 groups of classes (severity levels) from $\mathcal{Y}_{OOD}$, with increasing severities, such that severity level $i$ is the $i^{th}$ percentile of all severities. We further expand on how we chose the severity levels, their statistical meaning and the construction of the multiple C-OOD datasets for our experiments in Appendix L.

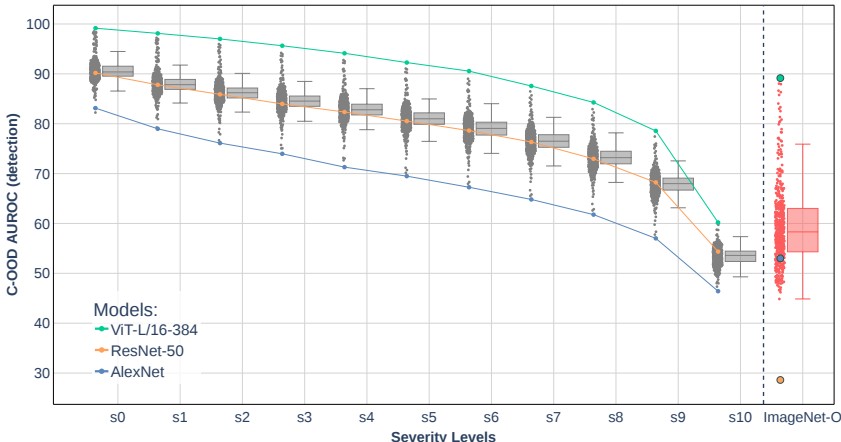

Figure 6: OOD performance across 11 severity levels. Note how the detection performance decreases for all models as we increase the difficulty until it reaches near chance detection performance at the last severity ($s_{10}$). The top curve belongs to ViT-L/16-384, which beats all models at every severity level. We also observe how the previous C-OOD benchmark, ImageNet-O does not reflect the true OOD performance of the models, since it was designed to specifically fool ResNet-50, and so it is more difficult for models similar to ResNet-50 than other models.

Figure 6 presents the C-OOD detection performance of 484 models across 11 C-OOD severity levels (recall that severity levels are constructed individually for each model). The box plots clearly show a monotone AUROC performance degradation. In addition, we see that ViT-L/16-384 is consistently the best model for each level (recall that this model is also the best for ID).

Most works on OOD detection use small scale datasets that generally do not resemble the training distribution and, therefore are, easy to detect. The use of such sets often causes C-OOD detectors to appear better than they truly are in harder tasks. Motivated by this deficiency, Hendrycks et al. (2021) introduced the ImageNet-O dataset as a solution. ImageNet-O, however, has two limitations. First, it lacks severity levels. Second, the original intent in the creation of ImageNet-O was to include only hard C-OOD instances. The definition of "OOD hardness", however, was carried out with respect to ResNet-50's difficulty in detecting OOD classes. This property makes ImageNet-O strongly biased. Indeed, the right-most box in Figure 6 corresponds to the performance of the 484 models over ImageNet-O. The orange dot in that box corresponds to ResNet-50, whose OOD detection perfromance is severely harmed by these data. Nevertheless, it is evident that numerous models perform quite well. In this respect, it can be argued that the proposed C-OOD dataset generator (see Appendix L) has better properties and is clearly, not biased toward a single architecture.

The next question we ask is does ID uncertainty estimation ranking performance indicate better C-OOD detection performance (and vice versa)? Figure 7 shows a scatter plot of ID vs. C-OOD AUROC performance of all the tested models. The overall Spearman correlation is 0.43. The legend indicates correlations obtained by specific families. For instance, ViTs are among the most

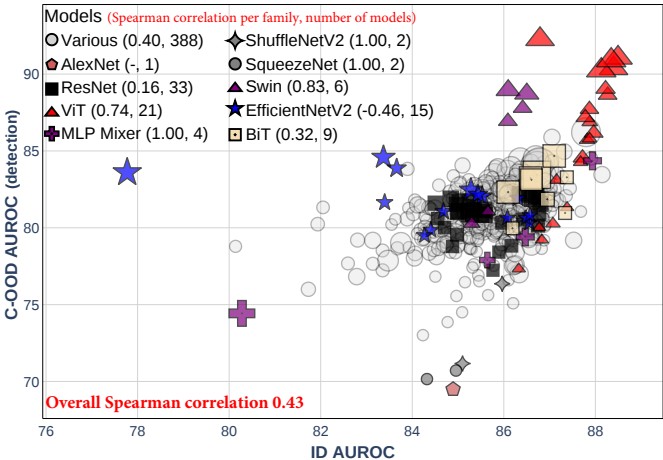

Figure 7: OOD detection performance in severity 5 vs. In-Distribution uncertainty estimation performance. We notice that the best performing network in one task is not the best in the other, but both belong to the same family, ViTs

correlated largest sample size family (0.74). This means that a ViT model is likely to perform well in both ID and C-OOD. Note that the worst performing models in C-OOD detection are small, optimized models. We further discuss correlations with other factors such as accuracy, model size, input size and embedding size in Appendix M.7.

Due to lack of space, a number of additional interesting observations and results are presented in Appendix M. We mention the most interesting ones here: (1) In accordance with the ID results (see Section 3), among all training regimes, distillation improves performance the most across all severities; see Figure 21 in Appendix M.3. (2) At the outset it could be anticipated that ImageNet21k pretraining will hinder C-OOD detection performance (due to its exposure to the OOD classes in training). Surprisingly, we observe that pretraining on ImageNet21k somewhat helps performance at severity levels up to level 6; see Figure 21 in Appendix M.3. (3) ViTs appear to achieve the best C-OOD detection performance per-model size (# parameters); see M.1. (4) Using entropy, as $\kappa$, improves C-OOD detection performance in most cases; see Appendix M.4. (5) The use of MC-dropout for C-OOD detection is investigated in Appendix M.5.

## 5    CONCLUDING REMARKS

We presented a comprehensive study of the effectiveness of numerous DNN architectures (families) in providing reliable uncertainty estimation, including the impact of various techniques on improving such capabilities. Moreover, we considered both in-distribution and novel (graded) class-out-of-distribution settings. Our study led to many discoveries and perhaps the most important ones are: (1) architectures trained with distillation almost always improve their uncertainty estimation performance, (2) temperature scaling is very useful not only for calibration but also for ranking and selective prediction, and (3) no other DNN (evaluated in this study) had ever demonstrated an uncertainty estimation performance comparable—in any metric tested or setting, in-distribution or class-out-of-distribution—to the ViT architecture.

Our work leaves open many interesting avenues for future research and we would like to mention a few. Perhaps the most interesting question is why distillation is so beneficial in boosting uncertainty estimation. Next, what is the architectural secret in vision transformers (ViT) that enables their uncertainty estimation supremacy. This question is even more puzzling given the fact that ViT supremacy is not shared with many other supposedly similar transformer-based models that we tested such as Touvron et al. (2021b; 2020); Liu et al. (2021); Han et al. (2021); Graham et al. (2021); d'Ascoli et al. (2021); Heo et al. (2021); Xu et al. (2021); El-Nouby et al. (2021); Zhang et al. (2021); Chu et al. (2021); Chen et al. (2021b). Finally, can we create specialized training regimes (e.g., Geifman & El-Yaniv (2019)), specialized augmentations, or even specialized neural architecture search (NAS) strategies that can promote superior uncertainty estimation performance?

## 6 REPRODUCIBILITY

The weights for all models evaluated in this paper are publicly available, and taken from the PyTorch and timm respositories (Paszke et al., 2019; Wightman, 2019), see Section 1. The algorithm for constructing our C-OOD dataset out of ImageNet21k, grouped into various severity levels, is detailed in Appendix L.

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

## A    THE INVESTMENT EXAMPLE

Let us consider two classification models for the stock market that predict whether a stock's value is about to increase, decrease or remains neutral (three-class classification). Suppose that Model A has a 95% true accuracy, and generates a confidence score of 0.95 on any prediction (even on missclassified instances); Model B has a 40% true accuracy, but always gives a confidence score of 0.6 on correct predictions, and 0.4 on incorrect ones. We now try and evaluate these two models with the uncertainty metrics mentioned in Section 1 to see which can reveal Model B's superior uncertainty estimation performance. AURC will fail due to its sensitivity to accuracy (the AURC of Model B is 0.12, more than twice as bad as the AURC for Model A, which is 0.05). NLL will rank Model A four times higher (Model A's NLL is 0.23 and Model B's is 0.93). The Brier score would also much prefer Model A (giving it a score of 0.096 while giving Model B a score of 0.54). Evaluating the models' calibration with ECE will also not reveal Model B's advantages, since it is less calibrated than Model A, which has perfect calibration (Model A has an ECE of 0, and Model B has a worse ECE of 0.4).

AUROC, on the other hand, would give Model B a perfect score of 1 and a terrible score of 0.5 to Model A. The selective risk for Model B would be better for any *coverage* of stock predictions below 40%, and for any SAC above 95% the coverage for Model A would be 0, but 0.4 for Model B.

Those two metrics are not perfect for any example. If instead we were to compare two different models for the task of predicting the weather, such that we cannot abstain from making predictions but are required to provide an accurate probabilistic uncertainty estimation of the model's predictions, AUROC and selective risk would be meaningless (due to the model's inability to abstain in this task), but ECE or the Brier Score would better evaluate the performance the new task requires.

## B    RANKING AND CALIBRATION VISUAL COMPARISON

A comparison of 484 models by their AUROC ($\times 100$, higher is better) and -log(ECE) (higher is better) on ImageNet is visualized in Figure 8. To compare models fairly by their size, we plot two graphs with the logarithm of the number of parameters as the X axis, so that models sharing the same x value can be compared solely based on their y value. In Figure 9 we set the X axis to be AUROC (higher is better), and see ViTs outperform any other architecture with a comparable amount of parameters by a large margin. We can also observe using distillation creates a consistent improvement in AUROC. In Figure 10 we set the X axis to be the negative logarithm of ECE (higher is better) and observe a very similar trend, with ViT outperforming its competition for any model size.

## C    DEMONSTRATION OF E-AURC'S DEPENDENCE ON THE MODEL'S ACCURACY

*Excess-AURC* (E-AURC) was suggested by Geifman et al. (2018) as an alternative to AURC (explained in Section 2). To calculate E-AURC, two AURC scores need to be calculated: (1) $AURC(model)$, the AURC value of the actual model and (2) $AURC(model^*)$, the AURC value of a hypothetical model with identical predicted labels as the first model, but that outputs confidence values that induce a perfect partial order on the instances in terms of their correctness. The latter means that all incorrectly predicted instances are assigned confidence values lower than the correctly predicted instances.

E-AURC is then defined as $AURC(model) - AURC(model^*)$. In essence, this metrics acknowledges that given a model's accuracy, the area of $AURC(model^*)$ is always unavoidable no matter how good the partial order is, but anything above that could have been minimized if the $\kappa$ function was better at ranking, assigning correct instances higher values than incorrect ones and inducing a better partial order over the instances.

This metric indeed helps to reduce some of the sensitivity to accuracy suffered by AURC, and for the example presented in Section 1, E-AURC would have given a perfect score of 0 to the model inducing a perfect partial order by its confidence values (Model B). It is easy, however, to craft

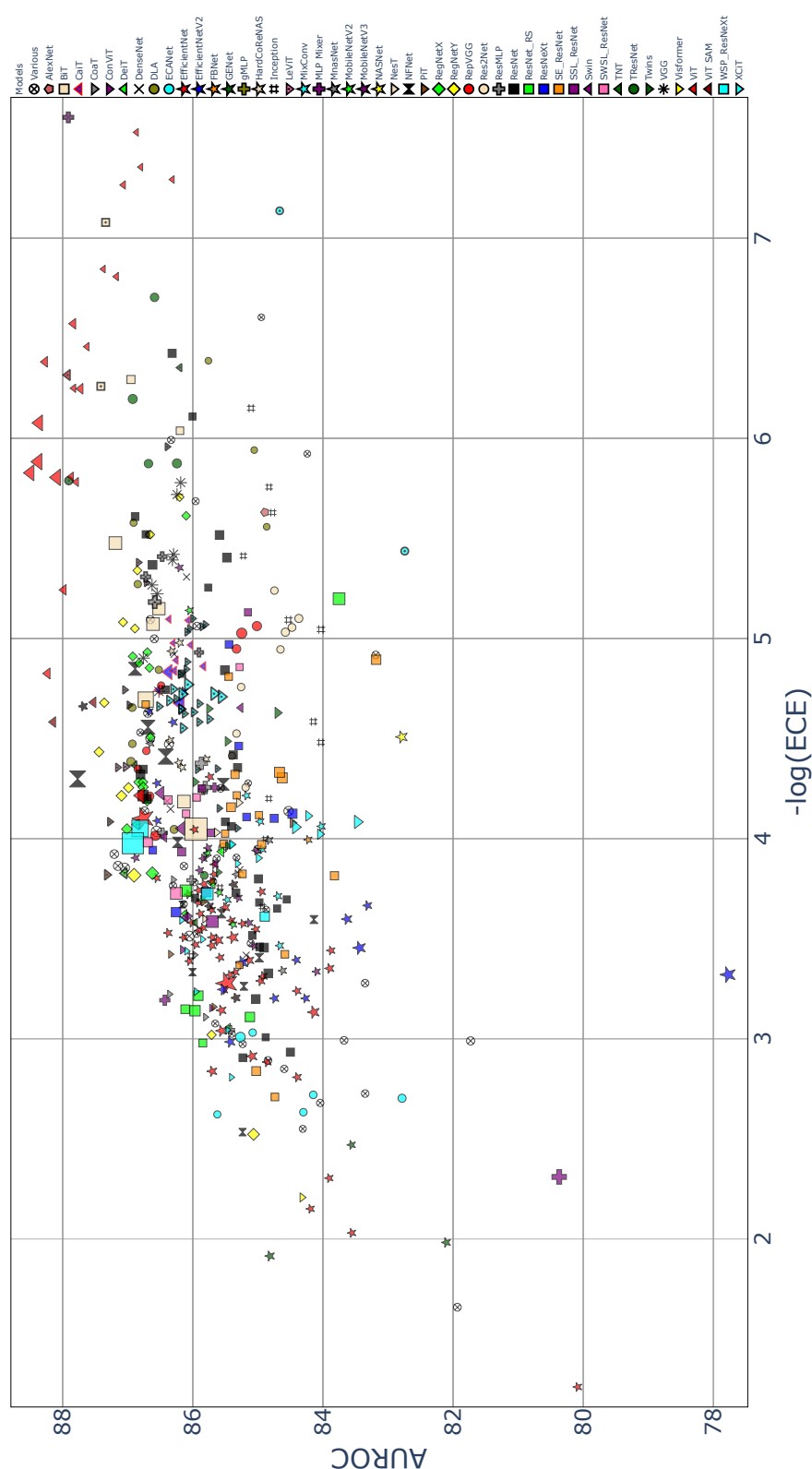

Figure 8: A comparison of 484 models by their AUROC (×100, higher is better) and log(ECE) (lower is better) on ImageNet. Each marker's size is determined by the model's number of parameters. Each dotted marker represents a distilled version of the original.

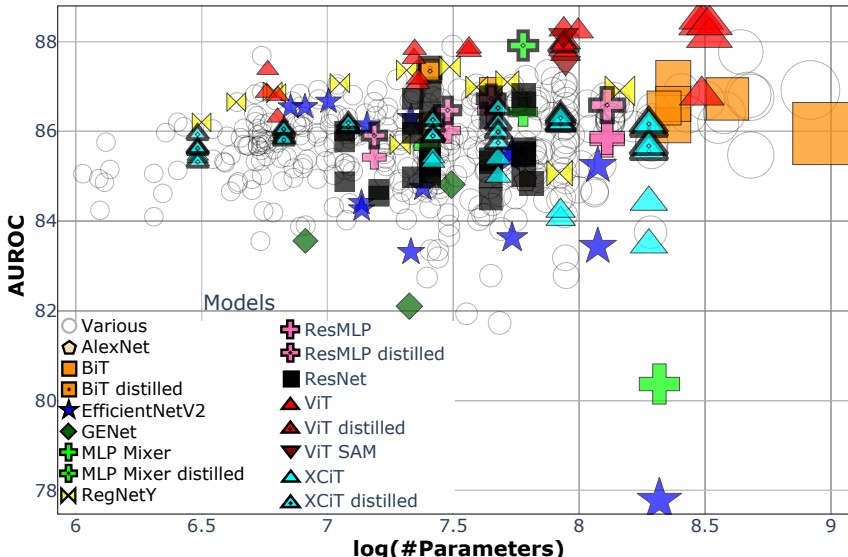

Figure 9: A comparison of 484 models by their AUROC (×100, higher is better) and log(number of model's parameters) on ImageNet. Each dotted marker represents a distilled version of the original.

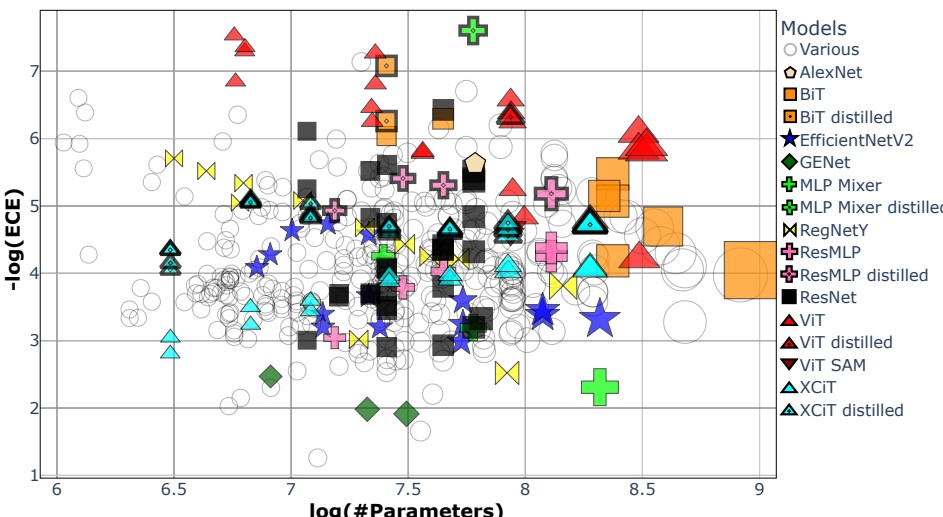

Figure 10: A comparison of 484 models by their -log(ECE) (higher is better) and log(number of model's parameters) on ImageNet. Each dotted marker represents a distilled version of the original.

examples showing that E-AURC prefers models with higher accuracy, even if they have lower or equal capacity to rank.

To demonstrate this in a simple way, let us consider two models with a complete lack of capacity to rank correct and incorrect predictions correctly, always outputting the same confidence score. Model A has an accuracy of 10% (thus an error rate of 90%), and Model B has an accuracy of 50%. A good ranking metric should evaluate them equally (the same way E-AURC gives the same score for two models that rank perfectly regardless of their accuracy). In Figure 11 we plot their RC curves, which are both straight lines due to their lack of ranking ability. We can calculate both of these models AURCs, $AURC(modelA) = 0.9, AURC(modelB) = 0.5$.

The next thing to calculate is the best AURC values those models could have achieved given the same accuracy if they had a perfect partial order. We plot these hypothetical models' RC curves in Figure 12. Their selective risk remains 0 for every coverage below their total accu-

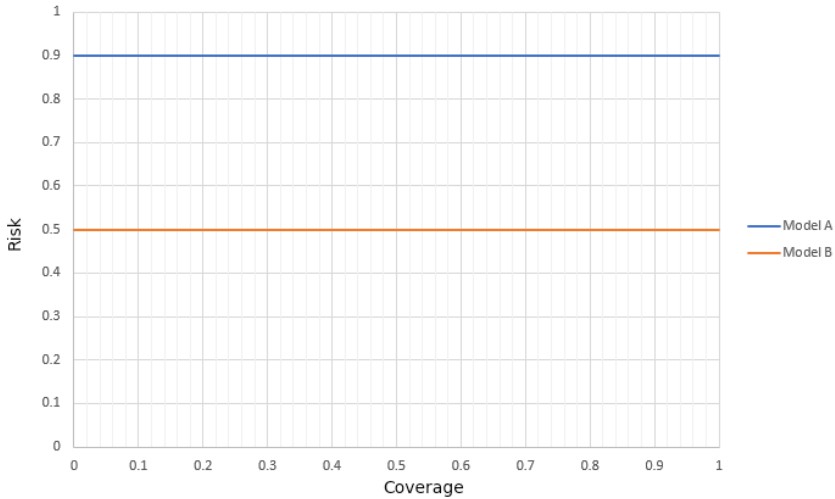

Figure 11: The RC curves for Models A and B.

racy, since these hypothetical models assigned the highest confidence to all of their correct in-stances first. As the coverage increases and they have no more correct instances to select, they begin to give instances that are incorrect, and thus their selective risk linearly deteriorates for higher coverages. Calculating both of these hypothetical models' AURCs gives us the follow-

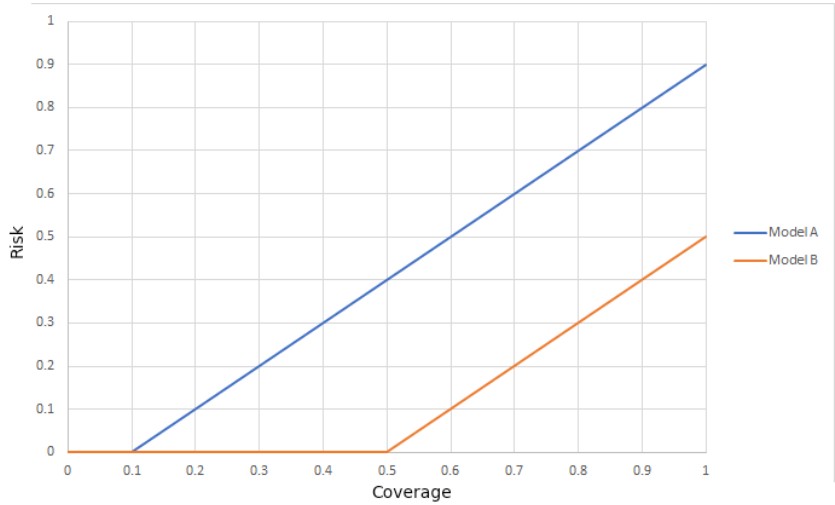

Figure 12: The RC curves for the hypothetically optimal versions of Models A and B.

ing: $AURC(modelA^*) = 0.405, AURC(modelB) = 0.125$. Subtracting our results we get: $E - AURC(modelA) = 0.9 - 0.405 = 0.495, E - AURC(modelB) = 0.5 - 0.125 = 0.375$ Hence, E-AURC prefers Model B over Model A, even though both do not discriminate at all between incorrect and correct instances.

## D    MORE ON THE DEFINITION OF RANKING

Let us consider a finite set $S_m = \{(x_i, y_i)\}_{i=1}^m \sim P_{X,Y}$. We assume that there are no two identical values given by $\kappa$ on $S_m$. Such an assumption is reasonable when choosing a continuous confidence signal.

We further denote $c$ as the number of concordant pairs (i.e., pairs in $S_m$ that satisfy the condition $[\kappa(x_i, \hat{y}|f) < \kappa(x_j, \hat{y}|f) \cap \ell(f(x_i), y_i) > \ell(f(x_j), y_j)])$ and $d$ as the number of discordant pairs (i.e., pairs in $S_m$ that satisfy the condition $[\kappa(x_i, \hat{y}|f) > \kappa(x_j, \hat{y}|f) \cap \ell(f(x_i), y_i) > \ell(f(x_j), y_j)]$

We assume, for now, that there are no two identical values given by $\ell$ on $S_m$. Accordingly, we can further develop Equation (1) from 2.1 using the definition of conditional probability,

$$\mathbf{Pr}[\kappa(x_i, \hat{y}|f) < \kappa(x_j, \hat{y}|f)|\ell(f(x_i), y_i) > \ell(f(x_j), y_j)] =$$
$$\frac{\mathbf{Pr}[\kappa(x_i, \hat{y}|f) < \kappa(x_j, \hat{y}|f) \cap \ell(f(x_i), y_i) > \ell(f(x_j), y_j)]}{\mathbf{Pr}[\ell(f(x_i), y_i) > \ell(f(x_j), y_j)]},$$

which can be approximated empirically, using the most likelihood estimator, as

$$\frac{c}{\binom{m}{2}}. \tag{2}$$

We notice that the last equation is identical to Kendall's $\tau$ up to a linear transformation, which equals

$$\frac{c - d}{\binom{m}{2}} = \frac{c - d + c - c}{\binom{m}{2}}$$
$$= \frac{2c - (c + d)}{\binom{m}{2}} = \frac{2c}{\binom{m}{2}} - \frac{c + d}{\binom{m}{2}} =$$
$$2 \cdot \frac{c}{\binom{m}{2}} - 1 = 2 \cdot [\text{Equation 2}] - 1.$$

Otherwise, if the loss assigns two identical values to a pair of points in $S_m$, but $\kappa$ does not, then we get:

$$\frac{c}{c + d}. \tag{3}$$

which is identical to Goodman & Kruskal's $\gamma$-correlation up to a linear transformation

$$\frac{c - d}{c + d} = \frac{c - d + c - c}{c + d} = \frac{2c - (c + d)}{c + d} =$$
$$\frac{2c}{c + d} - \frac{c + d}{c + d} = 2 \cdot [\text{Equation 3}] - 1.$$

## D.1 INEQUALITIES OF THE DEFINITION

One might wonder why Equation (1) should have strict inequalities rather than non-strict ones to define ranking. As we discuss below, this would damage the definition:

(1) If the losses had a non-strict inequality:

$$\mathbf{Pr}[\kappa(x_1, \hat{y}|f) < \kappa(x_2, \hat{y}|f)|\ell(f(x_1), y_1) \geq \ell(f(x_2), y_2)]$$

Consequently, in the case of classification, for example, this probability would increase for any pairs consisting of correct instances with different confidences, which yields no benefit in ranking between incorrect and correct instances and motivates giving different confidence values for instances with the same loss—a fact that would not truly add any value.

(2) If the $\kappa$ values had a non-strict inequality:

$$\mathbf{Pr}[\kappa(x_1, \hat{y}|f) \leq \kappa(x_2, \hat{y}|f)|\ell(f(x_1), y_1) > \ell(f(x_2), y_2)].$$

This probability would increase for any pair $(x_1, x_2)$ such that $\kappa(x_1, \hat{y}|f) = \kappa(x_2, \hat{y}|f)$ and $\ell(f(x_1)) > \ell(f(x_2))$, although $\kappa$ should have ranked $x_1$ with a lower value. Furthermore, if a

$\kappa$ function were to assign the same confidence score to all $x \in \mathcal{X}$, then when there are no two identical values of losses, the definition's probability would be 1; otherwise, the more different values for losses there are, the larger it would grow. In classification with a 0/1 loss, for example, assigning the same confidence score to all instances would result in the probability being $Accuracy(f) \cdot (1 - Accuracy(f))$, which is largest when $Accuracy(f) = 0.5$.

## E  RANKING CAPACITY COMPARISON BETWEEN HUMANS AND NEURAL NETWORKS

In the field of metacognition, interestingly, the predictive value of confidence is evaluated by two different aspects: by its ability to *discriminate* between correct and incorrect predictions (also known as *resolution* in metacognition or ranking in our context) and by its ability to give well calibrated confidence estimations, not being over- or underconfident (Fiedler et al., 2019). These two aspects correspond perfectly with much of the research done in the deep learning field, with the nearly matching metric to AUROC of $\gamma$-correlation (see Section 2).

This allows us to compare how well humans rank predictions in various tasks and how models rank their own in others. Human AUROC measurements in various tasks (translated from $\gamma$-correlation) tends to range from 0.6 to 0.75 (Undorf & Bröder, 2019; Basile et al., 2018; Ackerman et al., 2016), but could vary, usually towards much lower values (Griffin et al., 2019). In our comprehensive evaluation on ImageNet, AUROC ranged from 0.77 to 0.88 (with the median value being 0.85), and in CIFAR-10 these measurements jump to the range of 0.92 to 0.94.

While such comparisons between neural networks and humans are somewhat unfair due to the great sensitivity required for the task, research that directly compares humans and machine learning algorithms on the same task exist. For example, in Ackerman et al. (2019), algorithms far surpass even the group of highest performing individuals in terms of ranking.

## F  CRITICISMS OF AUROC AS A RANKING METRIC

In this section we show why AUROC does not simply reward models for having lower accuracy, addressing such criticism. The paper by Ding et al. (2019) presented a semi-artificial experiment to demonstrate that AUROC might get larger the worse the model's accuracy becomes. They considerr a model $f$ and its $\kappa$ function evaluated on a classification test set $\mathcal{X}$, giving each a prediction $\hat{y}_f(x)$ and a confidence score $\kappa(x, \hat{y}_f(x)|f)$, which in this case is the model's softmax response. Let $\mathcal{X}^c = \{x^c \in \mathcal{X} | \hat{y}_f(x^c) = y(x)\}$ be the set of all instances correctly predicted by the model $f$, and define $x_{(i)}^c \in \mathcal{X}^c$ to be the correct instance that received the i-lowest confidence score from $\kappa$. Their example continues to consider an artificial model $f^m$ to be an exact clone of $f$ with the following modification: for every $i \leq m$, the model $f^m$ now predicts a different, incorrect label for $x_{(i)}^c$; however, its given confidence score remains identical: $\kappa(x_{(i)}^c, \hat{y}_f(x_{(i)}^c)|f) = \kappa(x_{(i)}^c, \hat{y}_{f^m}(x_{(i)}^c)|f^m)$. $f^0$ is exactly identical to $f$, by this definition, not changing any predictions. The paper shows how an artificially created model $f^m$ obtains a higher AUROC score the bigger its $m$. This happens even though "nothing" changed but a hit to the model's accuracy performance (by making mistakes on more instances).

First, to understand why this happens, we note that $f^1$: AUROC for $\kappa$ increases the more pairs of $[\kappa(x_1) < \kappa(x_2)|\hat{y}_f(x_1) \neq y(x_1), \hat{y}_f(x_2) = y(x_2)]$ there are. The model $f^1$ is now giving an incorrect classification to $x_{(1)}^c$, but this instance's position in the partial order induced by $\kappa$ has remained the same (since $\kappa(x_{(1)}^c)$ is unchanged); therefore, $|\mathcal{X}^c| - 1$ correctly ranked pairs were added: $[\kappa(x_{(1)}^c) < \kappa(x_{(i)}^c)|\hat{y}_f(x_{(1)}^c) \neq y(x_{(1)}^c), \hat{y}_f(x_{(i)}^c) = y(x_{(i)}^c)]$ for every $1 < i \leq |\mathcal{X}^c|$. Nevertheless, this does not guarantee an increase to AUROC by itself: if, previously, all pairs of (correct,incorrect) instances were ranked correctly by $\kappa$, AUROC would already be 1.0 for $f^0$ and would not change for $f^1$. If AUROC for $f^1$ is higher than it was for $f^0$, this means there exists at least one instance $x^w$ that was incorrectly predicted by the original model $f^0$ such that $\kappa(x_{(1)}^c) < \kappa(x^w)$. Every such *originally* wrongly ranked pair (by $f^0$) of $[\kappa(x_{(1)}^c) < \kappa(x^w)|\hat{y}_f(x^w) \neq y(x^w), \hat{y}_f(x_{(1)}^c) = y(x_{(1)}^c)]$ has been eliminated by $f^1$ wrongly predicting $x_{(1)}^c$. This, therefore, causes AUROC to increase at the expense of the model's accuracy.

Such an analysis neglects many factors, which is probably why such an effect is only likely to be observed in artificial models (and not among the actual models we have empirically tested):

1. It is unreasonable to assume that the confidence score given by $\kappa$ will remain exactly the same for an instance $x^c_{(i)}$ given it now has a different prediction. In the case of $\kappa$ being softmax, it assumes the model's logits have changed in a very precise and nontrivial manner. Additionally, by our broad definition of $\kappa$, which allows $\kappa$ to even be produced from an entirely different model than $f$, $\kappa$ receives the prediction and model as a given input (and cannot change or affect neither), and it is unlikely to assume changing its inputs will not change its output.

2. Suppose we find the setting reasonable and assume we can actually create a model $f^m$ as described. Let us observe a model $f^p$ such that $p = \min_m(\text{AUROC of } f^m\text{=1})$, meaning that $f^p$ ranks its predictions perfectly, unlike the original $f^0$. Is it really true that $f^p$ has no better uncertainty estimation than $f^0$? Model $f^p$ behaves very much like the investment "Model B" from our example in Section 1, possessing perfect knowledge of when it is wrong and when it is correct, allowing its users risk-free classification. So, given a model $f$, we can use the above process to produce an improved model $f^p$, and then we can even calibrate its $\kappa$ to output 0% for all instances below its threshold and 100% for all those above to produce a perfect model, which might have a small coverage but is correct every time, knows it and notifies its user when it truly knows the prediction. The increase in AUROC reflects such an improvement.

Not only do we disagree with such an analysis and its conclusions, but we also have vast empirical evidence to show that AUROC does not prefer lower accuracy models unless there is a good reason for it to do so, as we demonstrate in Figure 2 (comparing EfficientNet-V2-XL to ViT-B/32-SAM). In fact, out of the 484 models we tested, the model with the highest AUROC has also the $4^{th}$ highest accuracy of all models, and the overall Spearman correlation between AUROC and accuracy of all the models we tested is 0.03. Furthermore, Figure 2 also exemplifies why AURC, which was suggested by the mentioned paper as the alternative to AUROC, is a bad choice as a single number metric, and might lead us to deploy a model that has a worse selective risk for most coverages only due to its higher overall accuracy.

## G  AFFECTS OF THE MODEL'S ACCURACY, NUMBER OF PARAMETERS AND INPUT SIZE ON IN-DISTRIBUTION UNCERTAINTY ESTIMATION PERFORMANCE

Table 1 shows the relationship between uncertainty estimation performance and model's attributes and resources (accuracy, number of parameters and input size), measured by Spearman correlation. We measure uncertainty estimation performance by AUROC (higher is better) and -ECE (higher is better). Positive correlations indiciate good utilization of resources for uncertainty estimation (for example, a positive correlation between -ECE and the number of parameters indicates that as the number of parameters increases, the calibration improves). An interesting observation is that distillation can drastically change the correlation between a resource and the uncertainty estimation performance metrics. For example, undistilled XCiTs have a Spearman correlation of -0.79 between their number of parameters and AUROC, indicating that more parameters are correlated with lower ranking performance, while distilled XCiTs have a correlation of 0.35 between the two.

## H  KNOWLEDGE DISTILLATION EFFECTS ON IN-DISTRIBUTION UNCERTAINTY ESTIMATION

Figure 13 compares vanilla models to those incorporating KD into their training (represented by markers with thick borders and a dot). In a pruning scenario that included distillation, yellow markers indicate that the original model was also the teacher (Aflalo et al., 2020). While distillation using a different model tends to improve uncertainty estimation in both aspects, distillation by the model itself seems to improve only one—suggesting it is generally more beneficial to use a different model as a teacher. The fact that KD improves the model over its original form, however, is surprising, and

Table 1: The relationship between uncertainty estimation performance and the model's attributes and resources (accuracy, number of parameters and input size), measured by Spearman correlation. Positive correlations indicate good utilization of resources for uncertainty estimation.

| Architecture | AUROC & Accuracy | -ECE & Accuracy | AUROC & #Parameters | -ECE & #Parameters | AUROC & Input Size | -ECE & Input Size | # Models Evaluated |
|---|---|---|---|---|---|---|---|
| EfficientNet | -0.16 | -0.29 | -0.22 | -0.29 | -0.26 | -0.38 | 50 |
| ResNet | -0.28 | -0.22 | 0.16 | 0.03 | -0.40 | -0.44 | 33 |
| XCiT distilled | 0.60 | 0.09 | 0.35 | 0.02 | 0.51 | 0.12 | 28 |
| XCiT | -0.68 | 0.89 | -0.79 | 0.94 | - | - | 28 |
| ViT | 0.95 | -0.62 | 0.71 | -0.78 | 0.22 | -0.27 | 20 |
| SE_ResNet | -0.46 | -0.02 | -0.53 | 0.20 | -0.02 | -0.35 | 18 |
| EfficientNetV2 | -0.70 | -0.45 | -0.63 | -0.47 | -0.59 | -0.40 | 15 |
| NFNet | 0.56 | 0.78 | 0.63 | 0.81 | 0.48 | 0.60 | 13 |
| Inception | -0.29 | 0.09 | -0.43 | 0.30 | -0.08 | 0.23 | 13 |
| RegNetY | -0.03 | -0.98 | 0.27 | -0.86 | - | - | 12 |
| RegNetX | 0.20 | -0.96 | 0.20 | -0.96 | - | - | 12 |
| CaiT distilled | 0.44 | -0.87 | 0.35 | -0.87 | 0.58 | -0.50 | 10 |
| DLA | 0.64 | -0.90 | 0.77 | -0.90 | - | - | 10 |
| MobileNetV3 | 0.37 | 0.59 | 0.42 | 0.60 | - | - | 10 |
| Res2Net | -0.70 | 0.27 | -0.68 | 0.60 | - | - | 9 |
| VGG | 0.81 | -0.98 | 0.71 | -0.90 | - | - | 8 |
| RepVGG | -0.71 | 0.50 | -0.57 | 0.21 | - | - | 8 |
| BiT | -0.33 | -0.81 | -0.20 | -0.85 | -0.46 | -0.25 | 8 |
| ResNeXt | -0.96 | 0.39 | -0.22 | -0.30 | - | - | 7 |
| ResNet RS | 0.00 | 0.79 | -0.18 | 0.82 | -0.30 | 0.82 | 7 |
| MixConv | -0.11 | 0.89 | -0.24 | 0.86 | - | - | 7 |
| DenseNet | 0.43 | -0.14 | 0.72 | 0.12 | - | - | 6 |
| HardCoReNAS | -0.60 | 0.26 | -0.49 | 0.37 | - | - | 6 |
| Swin | 0.71 | 0.14 | 0.77 | 0.26 | 0.41 | 0.00 | 6 |
| ECANet | -0.20 | 0.60 | -0.43 | 0.37 | 0.83 | 0.37 | 6 |
| Twins | -0.26 | 0.94 | -0.14 | 0.89 | - | - | 6 |
| SWSL ResNet | 0.94 | -0.89 | 0.77 | -0.83 | - | - | 6 |
| GENet | 0.50 | -1.00 | 0.50 | -1.00 | 0.87 | -0.87 | 6 |
| SSL ResNet | 0.14 | -1.00 | 0.26 | -0.94 | - | - | 6 |
| TResNet | 0.10 | -0.30 | 0.53 | 0.53 | -0.58 | -0.87 | 5 |
| CoaT | -0.10 | 0.90 | -0.10 | 0.50 | - | - | 5 |
| LeViT distilled | 0.60 | -0.90 | 0.60 | -0.90 | - | - | 5 |
| ResMLP | 0.20 | 1.00 | 0.15 | 0.97 | - | - | 5 |
| MobileNetV2 | -0.30 | 0.00 | -0.21 | 0.10 | - | - | 5 |
| PiT distilled | 1.00 | -1.00 | 1.00 | -1.00 | - | - | 4 |
| PiT | -0.40 | 1.00 | -0.40 | 1.00 | - | - | 4 |
| WSP ResNeXt | 1.00 | 0.80 | 1.00 | 0.80 | - | - | 4 |
| ResMLP distilled | 0.80 | 0.20 | 0.80 | 0.20 | - | - | 4 |
| MnasNet | 0.40 | 0.20 | 0.63 | 0.95 | - | - | 4 |
| DeiT distilled | 0.80 | -1.00 | 0.80 | -1.00 | 0.77 | -0.77 | 4 |
| DeiT | 0.40 | 0.80 | 0.40 | 0.80 | 0.26 | 0.26 | 4 |

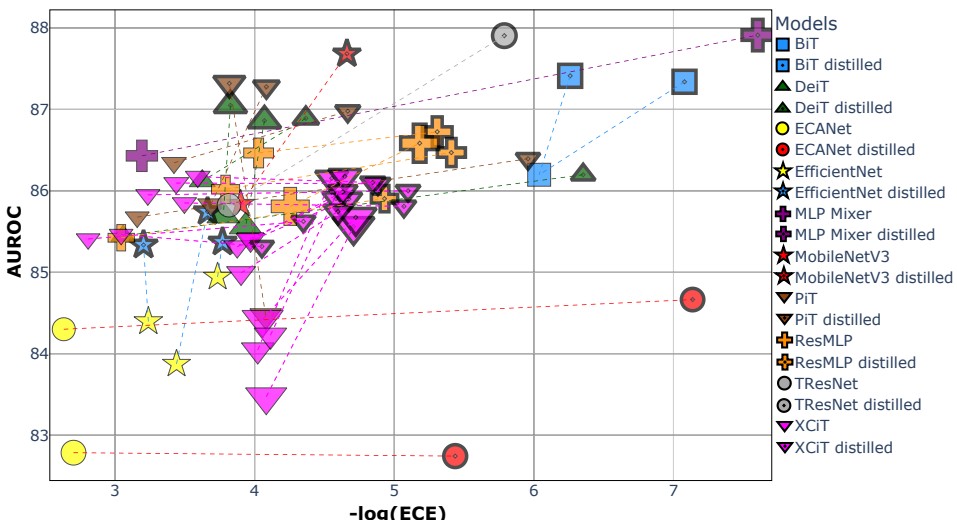

Figure 13: Comparing vanilla models to those incorporating KD into their training (represented by markers with thick borders and a dot). In a pruning scenario that included distillation, yellow markers indicate that the original model was also the teacher. The performance of each model is measured in AUROC (higher is better) and -log(ECE) (higher is better).

suggests the distillation process itself helps uncertainty estimation. Note that although this specific method involves pruning, evaluations of models pruned without incorporating distillation (Frankle & Carbin, 2018) revealed no improvement.

Moreover, it seems that the teacher does not have to be good in uncertainty estimation itself; Figure 14 shows this by comparing the teacher architecture and the students in each case.

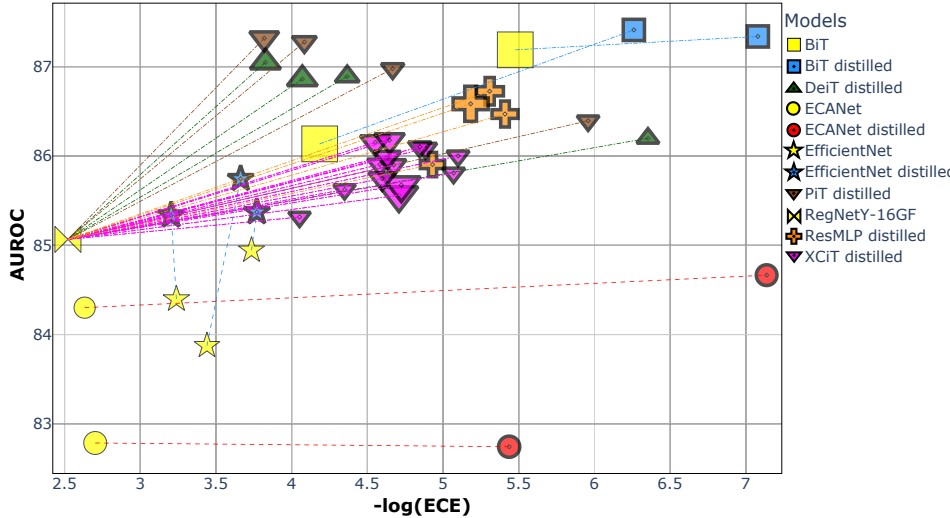

Figure 14: Comparing teacher models (yellow markers) to their KD students (represented by markers with thick borders and a dot). The performance of each model is measured in AUROC (higher is better) and -log(ECE) (higher is better).

While the training method by Ridnik et al. (2021) included pretraining on ImageNet-21k and demonstrated impressive improvements, comparison of models that were pretrained on ImageNet21k (Tan & Le, 2021; Touvron et al., 2021a) with identical models that were not pretrained showed no clear improvement in ECE, and, in fact, exhibit a degradation of AUROC (see Figures 3 and 4 in Section 3). This suggests that pretraining alone does not improve uncertainty estimation.

## I   MORE INFORMATION ABOUT TEMPERATURE SCALING

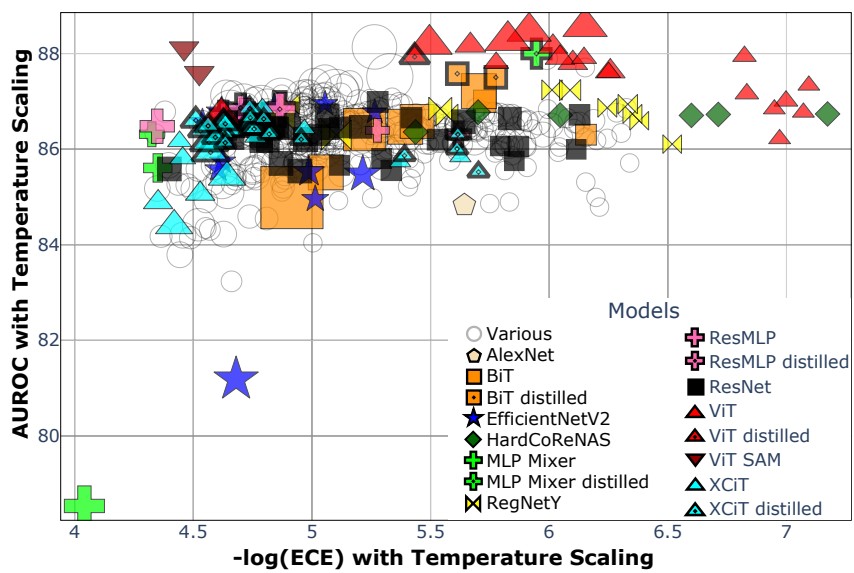

Figure 15: A comparison of 484 models after being calibrated with TS, evaluated by their AUROC (×100, higher is better) and -log(ECE) (higher is better) on ImageNet. Each marker's size is determined by the model's number of parameters. ViT models are still among the best performing architectures for all aspects of uncertainty estimation.

In Figure 15 we see how temperature scaling (TS) affects the overall ranking of models in terms of AUROC and ECE. While the ranking between the different architecture remains similar, the poorly performing models are much improved and minimize the gap between them and the best models. One particularly notable exception is HardCoRe-NAS (Nayman et al., 2021), with its lowest latency versions becoming the top performers in terms of ECE. In addition, models that benefit from TS in

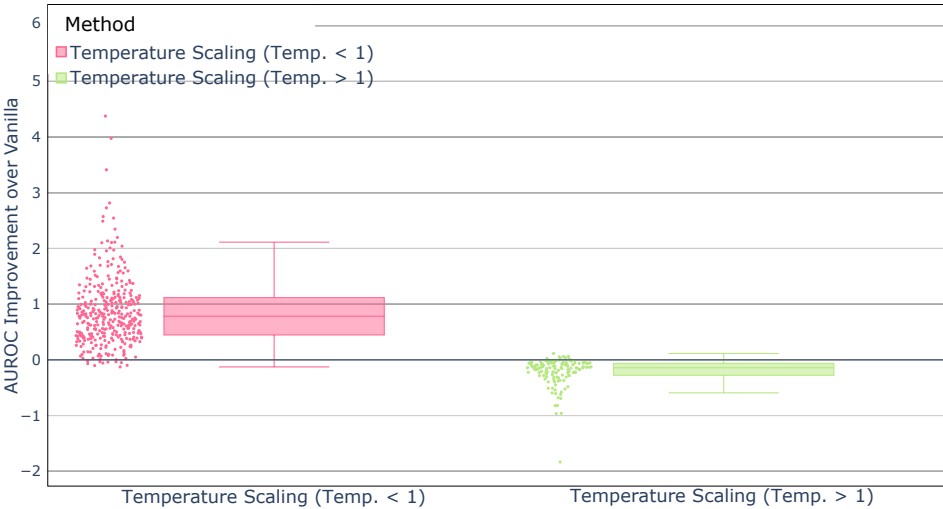

Figure 16: Out of 484 models evaluated, models that were assigned a temperature higher than 1 by the calibration process tended to degrade in AUROC performance rather than improve. Markers above the x axis represent models that benefited from TS, and vice versa.

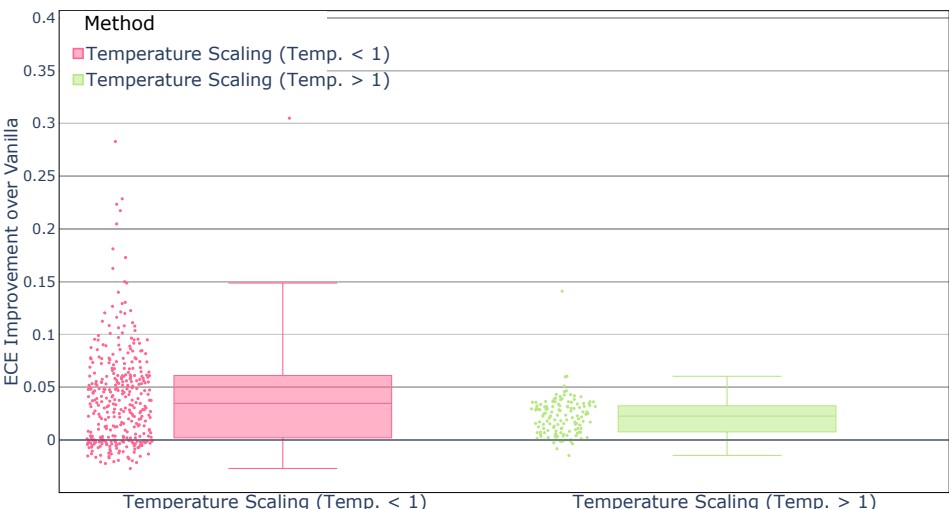

Figure 17: The relationship between temperature and the success of TS, unlike the case for AUROC, seems unrelated.

terms of AUROC tend to have been assigned a temperature lower than 1 by the calibration process (see Figure 16). The same, however, does not hold true for ECE (see Figure 17). This example also emphasizes the fact that models benefitting from TS in terms of AUROC do not necessarily benefit in terms of ECE, and vice versa. Therefore, determining whether to calibrate the deployed model with TS is, unfortunately, a task-specific decision.

We conduct TS as was suggested in Guo et al. (2017). For each model we take a random stratified sampling of 5,000 instances from the ImageNet validation set to calibrate on, and reserve the remainder 45,000 instances for testing. Using the box-constrained L-BFGS (Limited-Memory

Broyden-Fletcher-Goldfarb-Shanno) algorithm, we optimize for 5,000 iterations (though fewer iterations usually converge into the same temperature parameter) using a learning rate of 0.01.

## J   ARCHITECTURE CHOICE FOR PRACTICAL DEPLOYMENT BASED ON SELECTIVE PERFORMANCE

As discussed in Section 2, when we know the coverage or risk we require for deployment, the most direct metric to check is which model obtains the best risk for the coverage required (selective risk), or which model gets the largest coverage for the accuracy constraint (SAC). While each deployment scenario specifies its own constraints, for demonstration purposes we consider a scenario in which misclassifications are by far more costly than abstaining from giving correct predictions. An example for this could be classifying a huge unlabeled dataset (or for cleaning bad labels from a labeled dataset). While it is desirable to assign labels to a larger portion of the dataset (or to correct more of the wrong labels), it is crucial that these labels are as accurate as possible (or that correctly labeled instances are not replaced with a bad label).

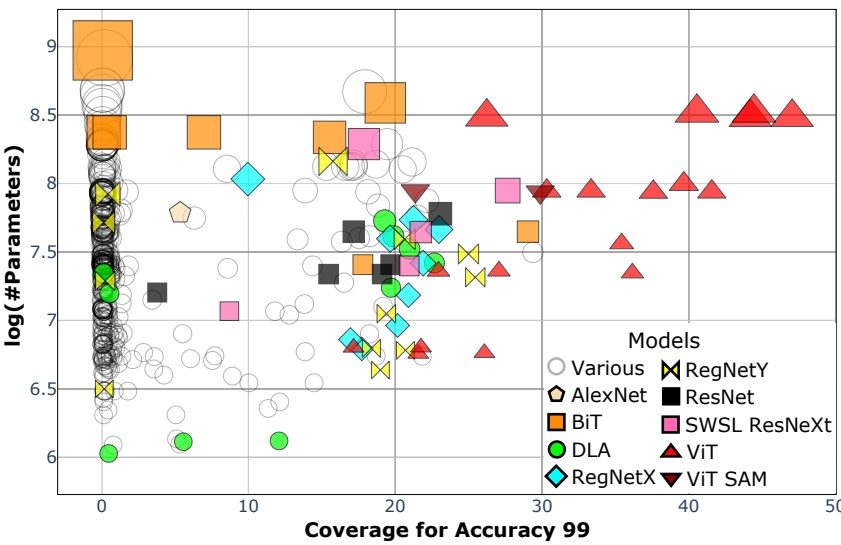

Figure 18: A comparison of 484 models by their log(number of model's parameters) and the coverage they are able to provide for a SAC of 99% (higher is better) on ImageNet.

To explore such a scenario, we evaluate all models on ImageNet to see which ones give us the largest coverage for a required accuracy of 99%. In Figure 5, Section 3 (paper's main body) we observe that of all the models studied, only ViT models are able to provide coverage beyond 30% for such an extreme constraint. Moreover, we note that the coverage they provide is significantly larger than that given by models with comparable accuracy or size, and that ViT models that provide similar coverage to their counterparts do so with less overall accuracy.

In Figure 18 we see that not only do ViT models provide more coverage than any other model, but they are also able to do so in any size category. To compare models fairly by their size, we present Figure 18, which sets the Y axis to be the logarithm of the number of parameters, so that models sharing the same y value can be compared solely based on their x value—which is the coverage they provide for a SAC of 99%. We see that ViT models provide a larger coverage even when compared with models of a similar size.

## K   EVALUATIONS OF MONTE-CARLO DROPOUT IN-DISTRIBUTION RANKING PERFORMANCE

MC-Dropout (Gal & Ghahramani, 2016) is computed using several dropout-enabled forward passes to produce uncertainty estimates. In classification, the mean softmax score of these passes is calcu-

Table 2: Comparing using MC-Dropout to softmax-response (vanilla).

| Architecture | Method | Accuracy | AUROC |
|---|---|---|---|
| MobileNetV3 Large | Vanilla | **74.04** | **86.88** |
| | MC-Dropout | 74 | 86.14 |
| MobileNetV3 Small | Vanilla | **67.67** | **86.2** |
| | MC-Dropout | 67.55 | 84.54 |
| MobileNetV2 | Vanilla | **71.88** | **86.05** |
| | MC-Dropout | 71.81 | 84.68 |
| VGG11 | Vanilla | **70.37** | **86.31** |
| | MC-Dropout | 70.21 | 84.3 |
| VGG11 (no BatchNorm) | Vanilla | **69.02** | **86.19** |
| | MC-Dropout | 68.95 | 83.94 |
| VGG13 | Vanilla | **71.59** | **86.3** |
| | MC-Dropout | 71.43 | 84.37 |
| VGG13 (no BatchNorm) | Vanilla | **69.93** | **86.24** |
| | MC-Dropout | 69.71 | 84.3 |
| VGG16 | Vanilla | **73.36** | **86.76** |
| | MC-Dropout | 73.33 | 85.02 |
| VGG16 (no BatchNorm) | Vanilla | **71.59** | **86.63** |
| | MC-Dropout | 71.47 | 84.97 |
| VGG19 | Vanilla | **74.22** | **86.52** |
| | MC-Dropout | 74.17 | 85.06 |
| VGG19 (no BatchNorm) | Vanilla | **72.38** | **86.55** |
| | MC-Dropout | 72.37 | 84.99 |

lated, and then a predictive entropy score is used as the final uncertainty estimate. In our evaluations, we use 30 dropout-enabled forward passes. We do not measure MC-Dropout's effect on ECE since entropy scores do not reside in $[0, 1]$.

We test this technique using MobileNetV3 (Howard et al., 2019), MobileNetv2 (Sandler et al., 2019) and VGG (Simonyan & Zisserman, 2014), all trained on ImageNet and taken from the PyTorch repository (Paszke et al., 2019).

The results comparing these models with and without using MC-Dropout are provided in Table 2.

The table shows that using MC-Dropout causes a consistent drop in both AUROC and selective performance compared with using the same models with softmax as the $\kappa$. These results are also visualized in comparison to other methods in Figure 3 in Section 3. MC-Dropout underperformance was also previously observed in (Geifman & El-Yaniv, 2017).

## L CONSTRUCTING C-OOD DATASET PER MODEL

Given model $f$, confidence function $\kappa$ and a large set of classes $\mathcal{Y}_{OOD}$ (in our case, ImageNet-21K), our goal is to build multiple datasets from $\mathcal{Y}_{OOD}$ that can be used in evaluating the model's performance in C-OOD detection. For that we start by pre-processing $\mathcal{O}$.

### L.1 PRE-PROCESSING IMAGENET-21K

Since ImageNet-21K contains our ID dataset (ImageNet-1K), the first step will be to remove all 1K classes from ImageNet-21K. After that, as a cautionary step, we remove all classes that are hypernyms or hyponyms of classes in ImageNet-1K because they might be unfair to include as an OOD class. For example, ImageNet-1K contains the class "brown bear", and ImageNet-21K has the class "bear" which is a hypernym for "brown bear" so it would not be fair to include it in a C-OOD detection test. After cleaning trivial classes, we remove classes with a low number of samples; we set our threshold to 200 samples. For classes with more than 200 samples we randomly select 200 samples and remove the rest. At the end of this process we are left with 12697 classes containing 200 samples each. We divide the samples in each class $c^y$ into 150 'estimation' samples (denoted by $c^y_{est}$) and 50 'test' samples (denoted by $c^y_{test}$). The estimation samples will later be used to estimate

the difficulty of the class and the test samples will be used to check the C-OOD performance if the class was chosen.

## L.2    CONSTRUCTION

Having conducted the pre-processing, we continue to compute the 11 severity levels (C-OOD datasets) for the given model $f$ (with its given $\kappa$).

We define the severity score of class $y$ to $(f, \kappa)$ as follows:

$$s(y|f, \kappa) = \frac{1}{|c^y|} \sum_{x \in c^y_{est}} \kappa(x|f).$$

We also define the severity score for a given group of classes $g$ as:

$$s(g|f, \kappa) = \frac{1}{|g|} \sum_{y \in g} s(y|f, \kappa).$$

Such a definition of $s$ is intuitive because it assumes that samples that the model is highly confident of are hard for it to distinguish from ID samples. We choose the size of each C-OOD dataset to be the same as the size of the of the ID dataset, 1000 classes. The number of subgroups in $\mathcal{Y}_{OOD}$ of size 1000 is huge ($\binom{12697}{1000} = 3.5 \times 10^{1518}$ groups), so instead of going over every possible group of classes, we choose to sort the classes by their severity score and then use a sliding window of size 1000 to define resulting in 11698 groups of classes with increasing severities. This choice for reducing the number of considered groups of classes was chosen for its simplicity. Using $s(g|f, \kappa)$ we calculate the severity score of each group, and sort them accordingly. Finally we choose the 11 groups to be the groups that correspond to the percentiles $\{10 \cdot i\}_{i=0}^{i=10}$ in the sorted groups array. This way we build the C-OOD dataset of severity level $i$ from the test samples of classes in group $i$. This heuristic procedure for choosing groups allows us to interpret the severity levels with percentiles. For example, severity level 5 contains classes that have the median severity among the considered groups.

The reason for choosing the number of classes in each group to be the same as the number of classes in the ID dataset (1000 classes per group) is because we wanted the C-OOD dataset to be equal in size to the ID dataset.

## M    LIST OF C-OOD OBSERVATIONS

In this section we list additional findings regarding C-OOD detection.

### M.1    PER-SIZE COMPARISON

The scatter plot in Figure 19 shows the relationship between the # of architecture parameters and its C-OOD AUROC performance. Overall, there is a moderate Spearman correlation of 0.45 between #parameters and the C-OOD performance when considering all tested networks. When grouping the networks by architecture families, however, we see that some architectures have high correlation between their model size and their C-OOD AUROC. Architecture families that exhibit this behavior are, for example, ViTs, Swins, EffecientNetV2 and ResNets whose correlations are 0.91, 0.94, 0.89, and 0.79, respectively. Other families exhibit moderate correlations, e.g., EffecientNet(V1) with a 0.47 Spearman correlation. Some architectures, on the other hand, have strong negative correlation, e.g., Twins (Chu et al., 2021), NesT (Zhang et al., 2020) and Res2Net (Gao et al., 2021), whose correlations are -0.94,-1.0, and -0.85, respectively.

Additionally, we note that ViT models are also the best even when considering a model size limitation similar to ResNet-50 or ResNet-101.

### M.2    HIGH CORRELATION WITH ACCURACY

Similarly, the scatter plot in Figure 20 shows the relationship between the architecture validation accuracy and its C-OOD AUROC performance. Based on their apparent correlation, increasing

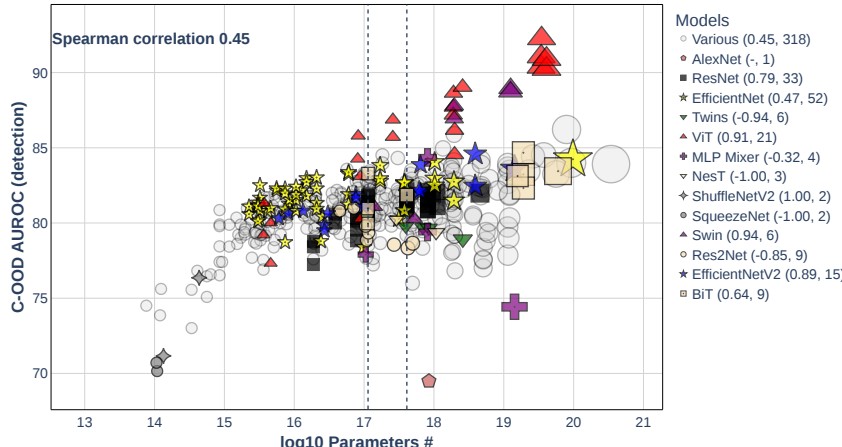

Figure 19: Number of architecture parameters vs. C-OOD AUROC performance at severity level 5 (median severity). The pair of numbers next to each architecture name at the legend correspond to its Spearman correlation and the number of models tested from that architecture (family), respectively. Note that ViT transformers are also the best when considering a model size limitation. Vertical lines indicate the sizes of ResNet-50 (left vertical line) and ResNet-101 (right vertical line).

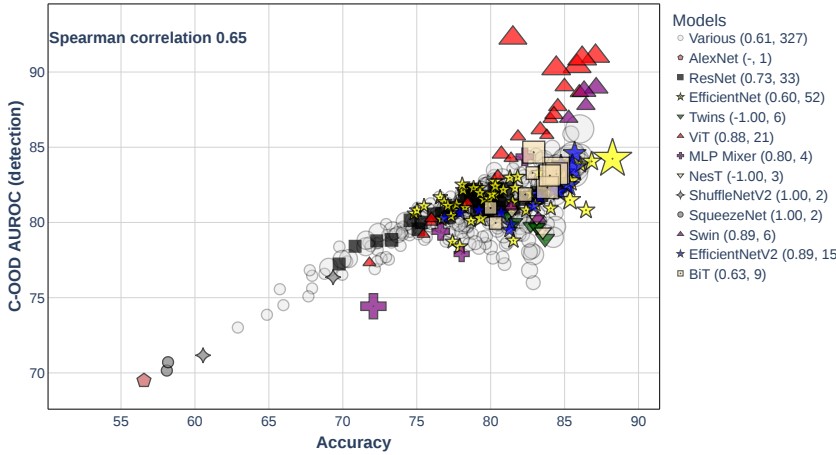

Figure 20: Architecture accuracy vs. C-OOD AUROC performance. In the legend, the pair of numbers next to each architecture name correspond to the Spearman correlation and the number of networks tested from that architecture (family), respectively. Accuracy appears to have a high correlation with the C-OOD detection performance, with a Spearman correlation of 0.65. Most architectures also hold this general trend except for Nest and Twins. Next to each architecture we report the Spearman correlation value and the number of networks tested from that architecture.

accuracy could indicate better C-OOD detection performance. When grouping the networks by architecture, we notice that most architectures also hold this trend. In Appendix M.7, we examine how the correlation between C-OOD AUROC and accuracy (among other metrics) change with severity.

## M.3 TRAINING REGIME EFFECTS

We evaluate the effect of several training regimes on C-OOD performance at various severity levels. The regimes we consider are: (1) Training that involves knowledge distillation in any form; (2) Adversarial training; (3) Pretraining on ImageNet21k; (4) Various forms of weakly or semi-supervised

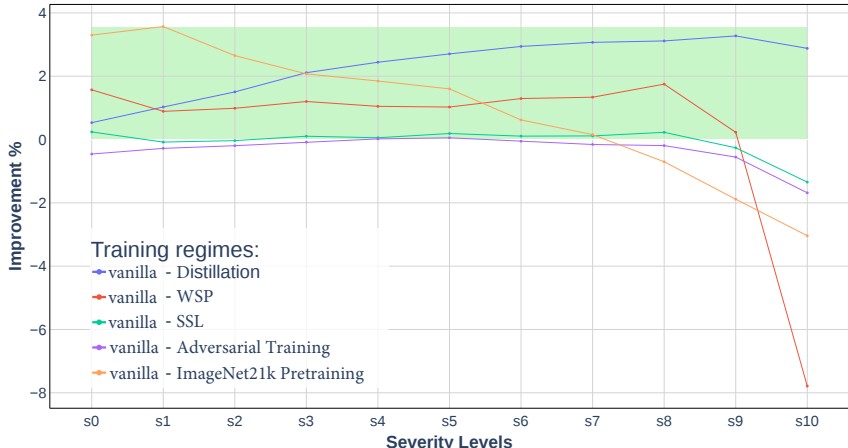

Figure 21: The average relative improvement when using distillation, pretraining, semi-supervised learning and adversarial training. The shaded green area indicates the area of positive improvement.

learning, including noisy student (Xie et al., 2020) and semi-supervised pretraining (Yalniz et al., 2019), which use extra unlabeled data; (5) Weakly supervised training (Mahajan et al., 2018a), which uses billions of weakly labeled Instagram images. The relative improvement generated by each one of the training regimes is depicted in Figure 21. We observe that distillation has a positive improvement on C-OOD detection performance on all severity levels, adversarial training has minimal effects on performance, and, somewhat surprisingly, training regimes that include pretraining on larger datasets (pretraining on ImageNet-21K and WSP) does not hinder performance in detecting OOD classes at lower severity levels, but degrades performance on higher severities—which is what we originally expected (due to exposure to the OOD classes in training).

## M.4 ENTROPY VS. SOFTMAX

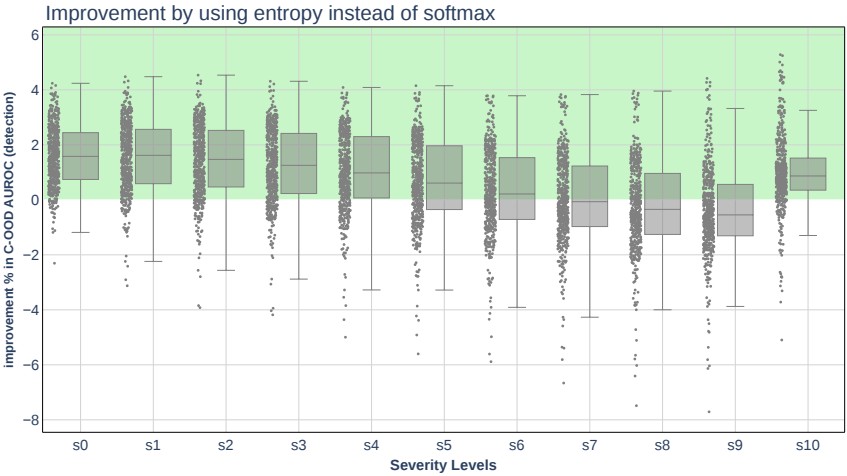

Figure 22: Relative improvement gain in C-OOD detection performance when using entropy instead of the softmax confidence signal. In median network terms, entropy offers positive improvement over softmax in most serverities except $s \in \{7, 8, 9\}$. The green shaded area indicates the area of positive improvement.

In our research we evaluated entropy as an alternative confidence rate signal ($\kappa$) for C-OOD detection.[1] For each network $f$ we re-run the algorithm described in Section L for extracting the classes

---

[1]Entropy is maximal when the distribution given by the network for $P(y|x)$ is uniform, which implies high uncertainty. To convert entropy into a *confidence signal* we use negative entropy.

for the 11 severity levels for $(f, \kappa_{entropy})$ and use them to benchmark the C-OOD detection for $(f, \kappa_{entropy})$ (Note that using the same C-OOD groups produced when using softmax might yield an unfair advantage to entropy). We compare the performance gain from switching to using entropy instead of the softmax score. The results are depicted using box-plots in Figure 22. We notice that in most cases using entropy improves the detection performance.

## M.5  MONTE-CARLO DROPOUT FOR C-OOD DETECTION

We evaluate MC-Dropout (Gal & Ghahramani, 2016) in the context of C-OOD detection. We use 30 dropout-enabled forward passes. The mean softmax score of these passes is calculated, and then a predictive entropy score is used as the final uncertainty estimate. We test this technique using MobileNetV3 (Howard et al., 2019), MobileNetv2 (Sandler et al., 2019) and VGG (Simonyan & Zisserman, 2014), all trained on ImageNet and taken from the PyTorch repository (Paszke et al., 2019).

For each of these three architectures we re-run the algorithm described in Section L to extract the classes for all 11 severity levels for $(f, \kappa_{\text{MC-dropout}})$ and use them to benchmark the C-OOD detection for $(f, \kappa_{\text{MC-dropout}})$.

The results are depicted using box-plots in Figure 23. We find that MC-Dropout improves performance at lower severity levels but degrades it at higher ones (with a few exceptions). We further analyze MC-dropout and recall that it is composed of two main components: (1) dropout-enabled forward passes (2) entropy of the mean probability vector from the forward passes. To test which component contributes the most to the perceived gains, we compare the C-OOD detection performance when using MC-dropout to the C-OOD detection performance when using entropy. We find that MC-dropout fails to improve upon entropy in most cases across all severity levels. This implies that MC-Dropout owes its C-OOD success to entropy and not to its forward passes. These results can be seen in Figure 24.

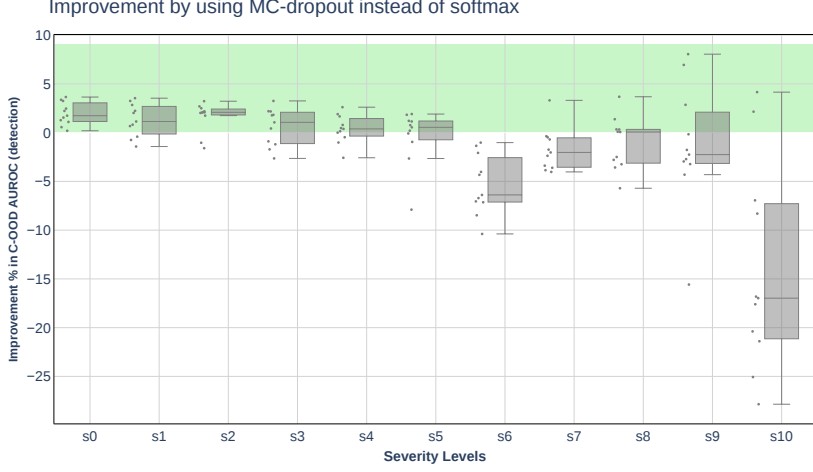

Figure 23: Relative improvement gain in C-OOD detection performance when using MC-Dropout instead of softmax confidence signal. We find that MC-dropout improves performance for most networks in severities up to $s_5$, and degrades performance for most networks in higher ones. Some outlier networks get a significant improvement when switching to MC-dropout at high severity levels.

## M.6  CORRELATION BETWEEN RANKINGS OF MULTIPLE SEVERITY LEVELS

Since we essentially have multiple benchmarks C-OOD datasets (i.e., the 11 severity levels) to test the performance models in C-OOD detection, and each severity level may rank the models differently, we now consider the question of how does these ranking change across severity levels. To this end we calculated the correlations between the rankings obtained at different severity levels. The resulting correlation matrix can be seen in Figure 25. Overall we observe high correlations,

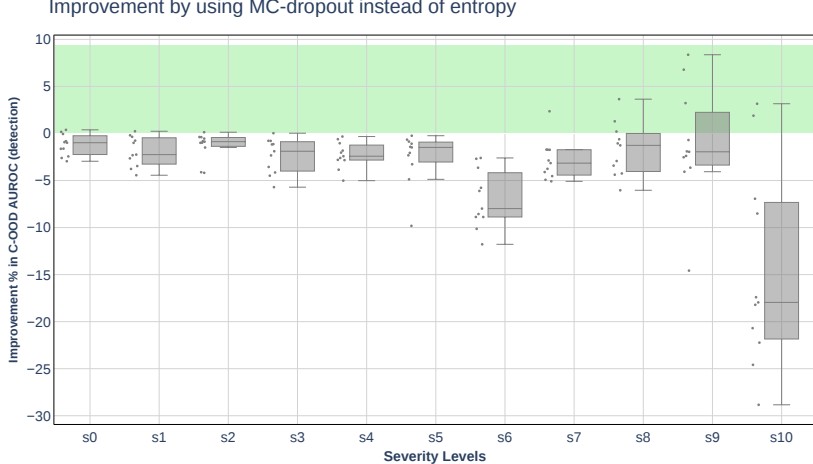

Figure 24: Relative improvement gain in C-OOD detection performance when using MC-dropout entropy confidence signal. We see that MC-dropout fails to improve upon entropy in most cases across all severity levels. This suggests that the main component in MC-dropout benefiting detection is the usage of entropy.

which means that different severity levels generally yield similar rankings of the models. We also notice that for each severity level $s_i$, the correlation with $s_j$ is higher the closer $j$ is to $i$. This is not surprising and might be anticipated because adjacent severity levels have close severity score by design.

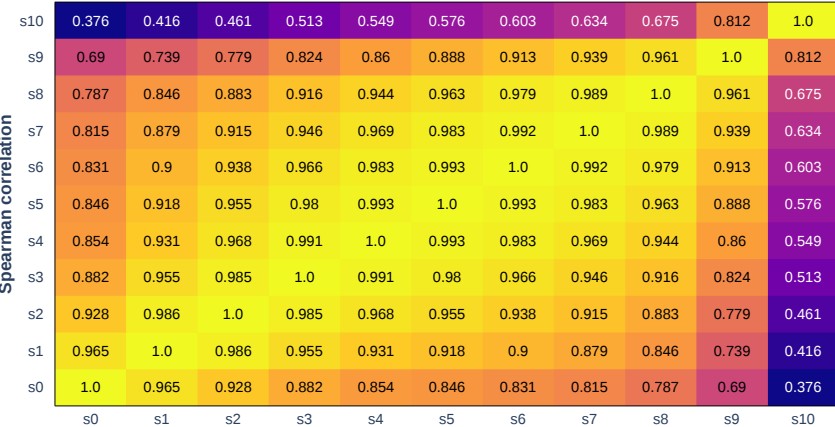

Figure 25: Spearman correlation between the rankings of the models given by different severity level.

## M.7 Correlations of Various Factors with C-OOD Detection Performance

We searched for factors that could be indicative or correlated with good performance in C-OOD detection. To this end we measure the correlations of various factors with the C-OOD detection AUROC performance across all severities. The results can be seen in the graphs of Figure 26. We observe that accuracy is typically a good indicator of the model's performance in C-OOD detection at most severity levels ($s_0 - s_8$), with Spearman correlation values in $[0.6, 0.7]$ at those levels. The next best indicative factors are the ID-AUROC performance, number of parameters, and the input image size (moderate correlations). Finally, the embedding size is only weakly correlated. Interestingly, ID-AUROC exhibits slightly increasing correlation up to severity $s_9$, and at $s_{10}$ becomes the most indicative factor for C-OOD detection. In contrast, all other investigated factors lose their indicative power at the highest severity levels ($s_9, s_{10}$).

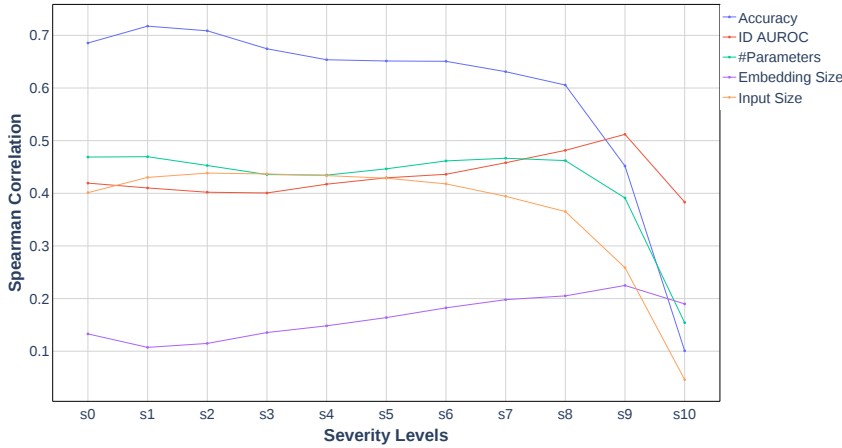

Figure 26: Spearman correlations between C-OOD detection AUROC and Accuracy, ID-AUROC, #Parameters, Input size, Embedding size across all severity levels.

## M.8 ANALYSING THE C-OOD CLASSES CHOSEN BY EACH MODEL

When analysing the classes chosen by the algorithm (described in Section L) for the various models and severity levels, we found that for each severity level, the union of the classes chosen for all models (for that severity level) spans all the classes in $\mathcal{Y}_{OOD}$ (namely, all the classes in ImageNet-21K after filtration). This implies that hard OOD classes for one model can easy for others.

In addition, for each severity level (except $s_{10}$) there is no single class that is shared between all models. However, in $s_{10}$ we find that there are some shared classes between all models. Moreover, there are specific instances (images) in those classes that obtain the same wrong prediction from many models; one interesting example for this shared wrong prediction of 105 models appears in Figure 27, where the butterfly image is wrongly classified as a fox squirrel.

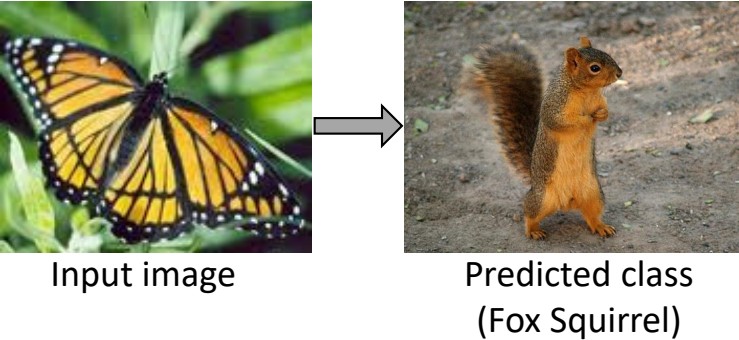

Figure 27: An instance of viceroy butterfly predicted to be a fox squirrel by 105 models.

