# OpenReview forum: "How to measure deep uncertainty estimation performance and which models are naturally better at providing it"
_ICLR.cc/2022/Conference — ICLR 2022 Submitted_

### Official Review · Reviewer_Wgpg · 2021-11-01

**Correctness:** 2
**Technical Novelty And Significance:** 2
**Empirical Novelty And Significance:** 2
**Recommendation:** 6
**Confidence:** 5

**Main Review:**

### Strength

1. The authors provided a thorough comparison of different architectures and training strategies.
2. The study compares 484 deep networks.
3. Their main strength is the observations on knowledge distillation and how providing extra information in any form either as pre-training or learning from unlabelled data helps in getting better uncertainty estimates.
4.  The authors compared different networks in OoD detection tasks of varying severity levels.
5.  They provide a comprehensive experimental design that considers OoD as found in real-applications that are not necessarily very far from training distribution.
6.  The authors demonstrate the strengths of the temperature scale (TS) method in proving ECE and uncertainty estimates.
7.  The paper demonstrates the strength of ViT architecture in providing better uncertainty estimates and predictive performance.


### Weakness
1.  A comparison with existing strategies of improving uncertainty estimates for a single-pass neural network is missing.
Many methods have been proposed which improves the uncertainty quantification for any network architecture. For examples

*  Mix-up learning: It is a training strategy to improve uncertainty estimates and model calibration [1].
* Strategies that impose a bi-Lipschitz constraint on feature extractors to learn a distance-aware feature representation for better OoD detection. A two-sided gradient penalty can be added to a network as an extra soft constraint [2].
*  Spectral normalization can be ​​combined with any architecture which has residual connections, resulting in better uncertainty estimates from that architecture [3].
* Adding extra regularization to learn an ordinal ranking relationship between different samples for better selective prediction [4].
* Temperature scaling as explored by the study

2. Experiments in Figure 3 and 4 show an improvement in ECE and AUC-ROC. This improvement over vanilla training came at a cost, such as the cost of training on unlabelled data as in semi-supervised learning.

3. Previous work, [4] has shown that Softmax entropy cannot capture epistemic uncertainty. Hence, the final output layer of the classifier, irrespective of the architecture, is usually not sufficient to capture epistemic uncertainty arising from OoD test samples.

4.  In section 4, the authors used the average confidence given by κ to identify OoD samples. Average confidence is a measure of softmax entropy and hence, is not sufficient to capture epistemic uncertainty.

5.  The figures 1, 5 and 7  are not much readable.

6.  In Figure 2, it’s not clear how to choose the sample Sm, so that we get different coverage. For example, most of the models report results on 100% coverage over the test sets of benchmark datasets. Should we mix some OoD samples in Sm so that we expect the model to have less than 100% coverage?  Or given any Sm, we partially rank samples based on confidence score, and then use different thresholds of confidence score to obtain a specific coverage level?
In this case, what if the model makes overconfident predictions, and hence confidence score is not sufficient to obtain the desired coverage level?


### Actions

1.  I would like to see quantitative comparison among
*  baseline models,
*  baseline model + [extra constraints for improved uncertainty [1-4],
*  Models with improved architecture

This quantitative comparison will help me understand what part of the improvement in uncertainty quantification can be achieved from a given baseline model (+ extra constraints), before adapting to a more complex architecture.

2. I would like to see extended results in Fig. 3 and 4 to include quantitative comparison among
* baseline models,
* baseline model + [extra constraints for improved uncertainty [1-4]],
* Baseline model with different KD methods.

This quantitative comparison will help me understand what part of the improvement in uncertainty quantification can be achieved from a given baseline model (+ extra constraints), before adding KD.

3. For OoD detection, I would like to see a comparison of
* baseline models such as ResNet
* Models with advance architectures (such as ViT)
*Feature-density based models that use data likelihood estimates to identify OoD samples.

This quantitative comparison will help me understand if advanced architectures reduce the need for extra feature-density based head such as gaussian mixture model to capture epistemic uncertainty.

4. For results in Fig.3 and 4, the authors should add analysis to quantify the added cost of knowledge distillation vs gain in ECE/Accuracy.
Report comparison on metrics such as extra training/pre-processing time.

5. I would like to see a through discussion on the results, summarizing the key findings in the form of clear guidelines:
* A guideline to follow when deciding for model architecture for improved uncertainty estimates.
* A guideline to consider when choosing different knowledge distillation (KD) methods for learning better classifiers. Do KD methods perform superior to baseline models [extra constraints for improved uncertainty]?

6. The authors should consider adding more text to explain the results in Figure 2.
7. The authors should consider adding a table to summarize the different architectures explored in the study.
Provide additional details on how different models with the same baseline architectures but different sizes or losses or knowledge distillation methods are considered. This table will greatly improve the readability of the current draft.


[1] On Mixup Training: Improved Calibration and Predictive Uncertainty for Deep Neural Networks

[2] Joost van Amersfoort, Lewis Smith, Yee Whye Teh, and Yarin Gal. Uncertainty estimation
using a single deep deterministic neural network. In International Conference on Machine
Learning, 2020.

[3] Jeremiah Zhe Liu, Zi Lin, Shreyas Padhy, Dustin Tran, Tania Bedrax-Weiss, and Balaji Lakshminarayanan. Simple and principled uncertainty estimation with deterministic deep learning via distance awareness. In Advances in Neural Information Processing Systems 33, 2020.

[4] Confidence-Aware Learning for Deep Neural Networks

[5] Deterministic Neural Networks with Inductive Biases Capture Epistemic and Aleatoric Uncertainty




**Summary Of The Paper:**

The paper provides an empirical comparison of uncertainty estimates obtained from 484 deep neural networks (DNN), trained for image classification tasks on the ImageNet dataset. They compared uncertainty estimation performance of different architectures and training strategies (knowledge distillation) on many quantitative metrics such as AUC-ROC on classification tasks, Area under the risk coverage curve (AURC), etc. The authors finally summarize their findings on what architectures are best at providing uncertainty estimations and how to compare different methods on OoD detection and uncertainty in in-distribution samples.


**Summary Of The Review:**

Overall the paper provides a thorough comparison of different networks on their uncertainty quantification. The paper largely did not consider the vast literature on improving uncertainty quantification of any given architecture.

---

> ### Author Response · Authors · 2021-11-11
> **Action items part1**
>
> Thank you for your detailed feedback and the proposed detailed action items,
>
> We liked your suggestion for including an evaluation of models trained using mix-up learning. In fact, following your remark we already made such evaluations. It does indeed improve uncertainty estimation for both the ID and C-OOD settings as you predicted. In ID, for example, the median improvement of AUROC is 0.49 (for TS, it is 0.51, and for KD, it is 0.75).
>
> As for the action items, a substantial amount of your items already partially appear in the paper (without the constraints):
>
> *"I would like to see quantitative comparison among
> baseline models,
> baseline model + [extra constraints for improved uncertainty [1-4],
> Models with improved architecture"*
>
> Figure 1 shows a comparison of all models, with the ViT models distinguished from the others. A full version of the figure (an interactive plotly) is attached as supplementary material for the reader's convenience.
>
> *"I would like to see extended results in Fig. 3 and 4 to include quantitative comparison among
> baseline models,
> baseline model + [extra constraints for improved uncertainty [1-4]],
> Baseline model with different KD methods."*
>
> We apologize for the misunderstanding: Figures 3 and 4 already show a quantitative comparison between baseline models and various training regimes (including KD). In the figure, each point indicates the difference between the same architecture with and without the mentioned method (e.g., KD). This will be made clear in the revision.
> Moreover, Figure 13 in Appendix H shows a more detailed comparison of KD trained models with their baselines.
>
> *"For OoD detection, I would like to see a comparison of
> baseline models such as ResNet
> Models with advance architectures (such as ViT)
> Feature-density based models that use data likelihood estimates to identify OoD samples."*
>
> Figure 6 includes such a comparison between ResNet50, a ViT and AlexNet (with all other 481 models also visible) across all severity levels of C-OOD (and on ImageNet-O, a competing OOD dataset for comparison).
> We will include an additional figure to make this comparison even clearer in the revision.
>
> *"A comparison with existing strategies of improving uncertainty estimates for a single-pass neural network is missing. Many methods have been proposed which improves the uncertainty quantification for any network architecture."*
>
> We agree that a comparison of existing methods specifically made for improving uncertainty estimates is important. The aim of our paper, however, as its title suggests, is to find out which models are better at providing it *naturally* (without any modifications). We went beyond our original objective and evaluated TS as well since our budget allowed it, and we were able to test multiple training regimes since many original pretrained weights were readily available for research. As a result, our paper has grown to be as large and filled with experiments and results as it is now.
> In addition to mix-up (for which we extracted results), running your requested constraints for 484 models for each setting would require a substantial amount of computations that we do not possess.
>
> However, we will do our best to implement as many of your requests as we can. As part of your third action item, we are currently extracting deep deterministic uncertainty (DDU) and mahalanobis distance data for the comparison.
>
> *"In section 4, the authors used the average confidence given by kappa to identify OoD samples. Average confidence is a measure of softmax entropy and hence, is not sufficient to capture epistemic uncertainty."*
>
> We chose the softmax as the primary kappa for our OOD detection experiments based on many OOD detection papers [1,2,3,4,5]. While we agree with you that softmax shouldn't be the best estimator for OOD detection, in this case we chose to focus on the baseline estimator that is *widely* accepted in the OOD detection literature. Moreover, our novel C-OOD benchmark dataset will hopefully encourage future research into finding the optimal OOD detection estimator for deep neural networks.
>
> [1] A Baseline for Detecting Misclassified and Out-of-Distribution Examples in Neural Networks
>
> [2] Confidence-based Out-of-Distribution Detection: A Comparative Study and Analysis
>
> [3] Natural Adversarial Examples
>
> [4] Enhancing The Reliability of Out-of-distribution Image Detection in Neural Networks
>
> [5] Out-of-Distribution Detection using Multiple Semantic Label Representations
>
> *"The figures 1, 5 and 7 are not much readable."*
>
> We will do our best to improve these figures. In the meantime, in the supplementary material, there is an interactive plotly and a full version of figure 1.

---

> > ### Author Response · Authors · 2021-11-11
> > **Action items part2**
> >
> > *"In Figure 2, it’s not clear how to choose the sample Sm, so that we get different coverage. For example, most of the models report results on 100\% coverage over the test sets of benchmark datasets. Should we mix some OoD samples in Sm so that we expect the model to have less than 100\% coverage? Or given any Sm, we partially rank samples based on confidence score, and then use different thresholds of confidence score to obtain a specific coverage level? In this case, what if the model makes overconfident predictions, and hence confidence score is not sufficient to obtain the desired coverage level?"*
> >
> > We will fix the explanation for this figure to make it clearer.
> > Figure 2 shows RC curves, which measure selective performance, in this context for an ID setting. In this example, Sm is the entire ImageNet validation set (no OOD instances included).
> >
> > *"given any Sm, we partially rank samples based on confidence score, and then use different thresholds of confidence score to obtain a specific coverage level?"*
> >
> > As you pointed out, this is correct.
> >
> > *"In this case, what if the model makes overconfident predictions, and hence confidence score is not sufficient to obtain the desired coverage level?"*
> >
> > Note that coverage refers to how many predictions the model hasn't rejected. Not all instances included in the coverage are correct. In fact, that is exactly why different models exhibit different selective performances: models that rank better would select fewer incorrect predictions into the required coverage.
> > According to Figure 2, for example, if the coverage is 0.2 (20\%), both ViT models (the blue and green curves) can select instances that are almost entirely correct (the risk is almost 0 for the blue curve and about 0.01 for the green one). When we look at EfficientNetV2 (red curve), however, we see that for the same coverage, it is only able to select a sample with 96\% accuracy. This example shows that even though the (red) EfficientNetV2 had many more correct instances to select from compared to the (green) ViT SAM - 12\% more correct instances (which is their difference in overall accuracy) - its inferior ranking performance causes it to select a sample with a worse risk.
> >
> > This figure serves two purposes:
> >
> > (1) Demonstrate that when rejection is possible, the overall accuracy is not a good predictor of performance after rejection.
> >
> > (2) Explain the drawbacks of using AURC as a ranking metric instead of AUROC, or examining the entire RC curve.
> >
> > *"For results in Fig.3 and 4, the authors should add analysis to quantify the added cost of knowledge distillation vs gain in ECE/Accuracy. Report comparison on metrics such as extra training/pre-processing time."*
> >
> > Unfortunately, since we didn't train them ourselves, we cannot specify the exact cost of their training.
> > In general, the KD methods we evaluated add the time it takes the selected teacher model to forward pass each batch, with some negligible addition to the student's backward pass (due to the increased computation for the loss function).
> > We emphasize that the primary focus of this paper is on which standard training methods excel at estimating uncertainty and not on how expensive they are to implement.
> >
> >
> > *"I would like to see a through discussion on the results, summarizing the key findings in the form of clear guidelines:
> > A guideline to follow when deciding for model architecture for improved uncertainty estimates."*
> >
> > The first guideline is simple: use ViT.
> > For any model size and for any type of uncertainty, as stated in the introduction and in Figures 9 and 10 (in the Appendix), ViTs outperform any other model by a wide margin.
> >
> > *"A guideline to consider when choosing different knowledge distillation (KD) methods for learning better classifiers. Do KD methods perform superior to baseline models [extra constraints for improved uncertainty]?"*
> >
> > As mentioned earlier, Figures 3 and 4 are comparative: each point represents the difference between an architecture trained with KD and the same architecture without it. This suggests that KD models are superior to baseline models. This is also true when comparing mix-up learning, as mentioned earlier.
> > There is no clear winner among the three KD methods we evaluated, and they all improve uncertainty estimation performance similarly.
> >
> > *"The authors should consider adding a table to summarize the different architectures explored in the study. Provide additional details on how different models with the same baseline architectures but different sizes or losses or knowledge distillation methods are considered. This table will greatly improve the readability of the current draft."*
> >
> > This is a good suggestion. We will create such a table.

---

> > ### Comment · Reviewer_Wgpg · 2021-11-13
> > **Extra constraints for improved uncertainty**
> >
> > After reading your comments, I understand that you are comparing off-the-shelf pre-trained models. If I understand correctly, your paper is a case study for someone who is looking for the best existing architecture with the most reliable uncertainty estimates. I understand, that the techniques that I have suggested, requires re-training from scratch hence are not directly applicable. However, from the current text, this idea is not converted. I will suggest re-writing part of the abstract and introduction to properly lay grounds for the scope of the study and its objective. Currently, it sounds like you are comparing a bunch of different models, but the motivation is missing. Also, I will suggest concluding with a clearly stated take-home message, highlighting your findings as guidelines, like which architecture to use for what benefits, what are the constraints (like computationally expensive) which may prevent someone from using the best model out there (ViT), how to find an alternative, and how to make whatever they have better (e.g., TS, specific types of KD).

---

> > > ### Author Response · Authors · 2021-11-13
> > > **Off-the-shelf pretrained models**
> > >
> > > Thank you for your reply,
> > >
> > > *"I understand, that the techniques that I have suggested, requires re-training from scratch hence are not directly applicable. However, from the current text, this idea is not converted. I will suggest re-writing part of the abstract and introduction to properly lay grounds for the scope of the study and its objective."*
> > >
> > > Our apologies for the confusion! In addition to re-writing the introduction to properly describe the scope, as you suggested, we will include the following clarification in the abstract:
> > >
> > > "We present a novel and comprehensive study of the uncertainty performance of 484 existing pretrained deep ImageNet classifiers that are available at popular repositories. Several interesting questions are addressed. For instance, out of these models, which one would provide the best performance in estimating uncertainty?"
> > >
> > >
> > > *"Also, I will suggest concluding with a clearly stated take-home message, highlighting your findings as guidelines, like which architecture to use for what benefits, what are the constraints (like computationally expensive) which may prevent someone from using the best model out there (ViT), how to find an alternative, and how to make whatever they have better (e.g., TS, specific types of KD)."*
> > >
> > > Thanks. We agree that the conclusions lack take-home guidelines at the moment, and this is a good suggestion. We will include these guidelines in the revision.

---

### Official Review · Reviewer_tT7p · 2021-11-02

**Correctness:** 4
**Technical Novelty And Significance:** 3
**Empirical Novelty And Significance:** 4
**Recommendation:** 8
**Confidence:** 3

**Main Review:**

I really loved this paper.
My only real criticism is that I felt like the paper could use some extra editing and structuring to make sure it has the impact it deserves to.
The abstract felt slightly too long and didn't draw focused enough attention to the main contributions of the paper, dwelling too much on details.
The introduction dives into specific trade-offs for various metrics before the metrics have been properly introduced. I would recommend moving the second half of the third paragraph and the fourth  and fifth paragraphs to a later section *after* the metrics have been defined.
I'd be tempted to slightly adjust the structure to: introduce the metrics; discuss the relative merits of the metrics; in-dist performance; introduce the novel OOD evaluation (which is a great contribution); ood performance; concluding remarks.

I really like the example in paragraph 2. I've not seen this made so explicit before and I think it does a great job motivating the work.

I also really like this paper as an example of how to make use of libraries of pre-trained models but nevertheless do extremely thorough empirical work.

Do you think the architecture-dependent correlations between ECE and AUROC are 'real'? Would this extend to other datasets? It makes me wonder if there's something unreliable going on there. In general it might be good to comment more on the limitation of relying on ImageNet, and add some consideration of how this might differ for other datasets/data modalities.

`both metrics are also' should be capitalized.

Title of Appendix G should be `Effects', not `Affects'

**Summary Of The Paper:**

The authors make use of 484 models pre-trained on ImageNet to assess both in- and out-of-distribution uncertainties along a number of metrics.
They not only provide a critical evaluation of the different metrics and their task-dependent strengths and weaknesses, but also discover previously unknown empirical patterns in different architectures and techniques.

**Summary Of The Review:**

This is a great paper. I have some quibbles about structure, but the basic results are extensive, interesting, and important. I would like to see this paper highlighted at the conference and would be disappointed if it were rejected (unless another reviewer points out to me prior work that I am not aware of).

---

> ### Author Response · Authors · 2021-11-11
> **Paper structure**
>
> We appreciate your positive feedback!
> We will definitely incorporate your suggestions for structural modifications in our revision.
>
> *"Do you think the architecture-dependent correlations between ECE and AUROC are 'real'? Would this extend to other datasets?"*
>
> The numbers speak for themselves, but we don't know. It's an intriguing question!
> We hypothesize these correlations are 'real' and strongly influenced by inductive bias factors embedded in the architecture itself. Some architectures are good at both calibration and ranking, while others will perform well only in calibration (or ranking). Of particular interest are the families exhibiting strong positive correlations, and what can we learn from them.
>
> Thank you for all of the other minor comments, we will fix them.

---

> > ### Comment · Reviewer_tT7p · 2021-11-25
> > **Thanks for your comments**
> >
> > I've read your responses to all the reviewers and I still like the paper. I think your paper is not a paper about epistemic uncertainty (because you use pre-trained deterministic models) and that's fine. It is also not a paper about OOD robustness techniques (because you use pre-trained deterministic models) and that's fine.
> >
> > Because many other people *also* use pre-trained deterministic models, your paper is still important (maybe make the contrast to Guo et al. 2017 "On Calibration..." more explicit).
> >
> > I notice, however, that you have not uploaded a restructured draft of the paper. Whether or not your paper is accepted, if you want it to have impact you will need to make the presentation clearer, otherwise readers will misunderstand its purpose. I am reducing my score by one notch, because I think I over-weighted my own enjoyment of the paper and underweighted its lack of clarity in my original review.

---

### Official Review · Reviewer_i3hp · 2021-11-02

**Correctness:** 4
**Technical Novelty And Significance:** 3
**Empirical Novelty And Significance:** 4
**Recommendation:** 6
**Confidence:** 3

**Main Review:**

The capability of models to reflect a calibrated uncertainty is interesting for different applications that require a reliable confidence indicator for the model predictions. However, some points regarding clarity on the document can be commented on.

The paper makes many references to figures, and sections of the appendix making the discussion of the work feel split between both documents. The appendix should be complementary and the paper should be self-explanatory, providing only complementary information like demonstrations or additional results. In this regard, it is recommended to review the information and results included in the main paper, compared with the content of the appendix, and include the necessary introduction to the concepts and definition, making it less for readers necessary to jump into the appendix and back to the paper.

More details about how the severity level can be included, even though the details are included in one section of the appendix (L), some introductory details can be added to the main paper.

Similarly, indicate some details about the models used, categories, criteria for selection.

**Summary Of The Paper:**

The paper presents an evaluation of different models in their capacity to reflect epistemic and aleatoric uncertainty and reviews different methods for uncertainty performance measurement. Models are classifiers trained on ImageNet. The number of models reported is 484.  The analysis also considers the cases when the data is in-distribution and out-of-distribution. The conclusions lead to a group of models (trained with distillation), improving uncertainty estimation performance, and the vision transformers architecture having the best uncertainty estimation performance.


**Summary Of The Review:**

In general, the document presents an interesting review about uncertainty estimating performance related to the models, and training strategies, however, perhaps due to space limitations, the main paper can present a general description of the work and rely on the appendix to give details. This can make the paper complicated to follow since in some cases, it might be necessary to jump into the appendix before continuing with the lecture of the main document.

---

> ### Author Response · Authors · 2021-11-11
> **Including the important information and results in the main paper**
>
> Thank you for your feedback!
>
> We are in agreement. The revision will include more of the main results and an introduction to severities, as much as possible.

---

### Official Review · Reviewer_4Gca · 2021-11-02

**Correctness:** 2
**Technical Novelty And Significance:** 2
**Empirical Novelty And Significance:** 2
**Recommendation:** 5
**Confidence:** 4

**Main Review:**

Positives:
- The paper focuses on uncertainty estimation in deep learning, which is an important research area that will enable risk aware and safe deployment of models.
- The paper conducts extensive experiments with 484 models to gather insights into training schemes and model architectures for capturing uncertainty.

Negatives:
- I overall disagree with the classification of uncertainty used in the paper. Uncertainty due to class-out-of-distribution should be grouped into epistemic uncertainty, which by definition describes the uncertainty stemming from lack of knowledge.
- Authors claim that ViT is a superior architecture for uncertainty quantification. I think this claim would have benefited by the distinction of aleatory and epistemic uncertainty. Aleatory uncertainty is related to the inherent randomness in data, and modeling choices do not affect it.
- Most of the analysis in the paper uses confidence score function \kappa to be the softmax values. This confidence score function will capture aleatory uncertainty but will not capture epistemic uncertainty. This distinction is important since there are evaluations performed with out of distribution data, which by definition stems from epistemic uncertainty. I believe the evaluations need to be consistent with these definitions.
- Distillation is observed to yield better uncertainty estimates. I think this result was not surprising given knowledge distillation's connections to ensembling, which is a state-of-art method for uncertainty quantification that captures epistemic uncertainty. I recommend adding a discussion on this point.
- Evaluations would benefit from reporting log likelihood numbers which is a common metric capturing the probabilistic distribution.

Minor comments:
- Typo in page 5. Use capital B in "both": "... tied values. both metrics are also closely .."
- Figure 4. For consistency, please report MC Dropout values here as well.
- Figure 6. It is interesting to comment on the slope of the curves, indicating how quickly the performance degrades with respect to severity levels.

**Summary Of The Paper:**

The paper presents an empirical study and focuses on evaluation of uncertainty with respect to different model architectures and training schemes. Using metrics capturing calibration and accuracy, the authors conclude that distillation based schemes are efficient in uncertainty estimation and ViT performs the best uncertainty estimates among the architectures considered.

**Summary Of The Review:**

Overall I recommend a rejection of the paper. I think the definitions of sources of uncertainty are important in terms of what we are able to quantify and those definitions need to align with the evaluations, and I believe there are some mismatches in this paper. I think this distinction will immensely help the empirical evaluations performed in this paper.

---

> ### Author Response · Authors · 2021-11-11
> **Epistemic uncertainty**
>
> Thank you for your feedback,
>
> *"I overall disagree with the classification of uncertainty used in the paper. Uncertainty due to class-out-of-distribution should be grouped into epistemic uncertainty, which by definition describes the uncertainty stemming from lack of knowledge."*
>
> We appreciate you bringing this to our attention. The C-OOD model is epistemic in nature (due to the fact that it cannot predict the unseen labels and lacks knowledge of them). We will fix our definitions throughout the paper.
>
> *"Most of the analysis in the paper uses confidence score function kappa to be the softmax values. This confidence score function will capture aleatory uncertainty but will not capture epistemic uncertainty. This distinction is important since there are evaluations performed with out of distribution data, which by definition stems from epistemic uncertainty. I believe the evaluations need to be consistent with these definitions."*
>
> We chose softmax as the primary kappa for our OOD detection experiments based on many OOD detection papers [1,2,3,4,5]. While we agree with you that softmax shouldn't be the best estimator for OOD detection, in this case, we chose to focus on the baseline estimator that is a *widely* accepted baseline in the OOD detection literature. Moreover, our novel C-OOD benchmark dataset will hopefully encourage future research into finding the optimal OOD detection estimator for deep neural networks.
>
> Having said that, we also ran experiments with kappa being predictive entropy, and you can see a short discussion of them in Appendix M.4 ("Entropy vs Softmax"). Here again, both distillation and ViTs were found to be the most successful when using predictive entropy for C-OOD.
>
>
> [1] A Baseline for Detecting Misclassified and Out-of-Distribution Examples in Neural Networks
>
> [2] Confidence-based Out-of-Distribution Detection: A Comparative Study and Analysis
>
> [3] Natural Adversarial Examples
>
> [4] Enhancing The Reliability of Out-of-distribution Image Detection in Neural Networks
>
> [5] Out-of-Distribution Detection using Multiple Semantic Label Representations
>
>
> *"Authors claim that ViT is a superior architecture for uncertainty quantification. I think this claim would have benefited by the distinction of aleatory and epistemic uncertainty. Aleatory uncertainty is related to the inherent randomness in data, and modeling choices do not affect it."*
>
> We are in agreement. In order to test this aspect, we plan to compare ViTs with a few baseline architectures, each tested using deep deterministic uncertainty (DDU) and Mahalanobis distance.
>
> *"Distillation is observed to yield better uncertainty estimates. I think this result was not surprising given knowledge distillation's connections to ensembling, which is a state-of-art method for uncertainty quantification that captures epistemic uncertainty. I recommend adding a discussion on this point."*
>
> We agree, thanks for the suggestion!
> The fact that knowledge distillation improves uncertainty estimation even when the teacher model has inferior uncertainty estimation performance, or when the teacher has the same weights as the student does, as discussed in Appendix H and visualized in Figure 14, is surprising and interesting to us.
>
> *"Evaluations would benefit from reporting log likelihood numbers which is a common metric capturing the probabilistic distribution."*
>
> We will include it in the paper.
>
> Thank you for the minor comments as well, we will fix them.

---

> > ### Comment · Reviewer_4Gca · 2021-11-29
> > **Thank you authors**
> >
> > I thank the authors for proposing to address my concerns on the classification of aleatory and epistemic uncertainty in the paper. I strongly believe that your changes will make a stronger paper, where the analysis is consistent with definitions of these key concepts. Therefore, I would like to increase my score by one point. However, I still believe the paper has room for improvements. I am afraid that I am not convinced that the paper offers novel contributions from a theoretical perspective. The empirical side of the paper is extensive but there are still concerns around selective prediction and kappa.

---

### Official Review · Reviewer_bFby · 2021-11-03

**Correctness:** 3
**Technical Novelty And Significance:** 1
**Empirical Novelty And Significance:** 3
**Recommendation:** 5
**Confidence:** 3

**Main Review:**

Strengthes:
- This work has extensive numerical studies on the metrics of uncertainty estimation performance and has drawn some observations.

Weaknesses:
- It is somewhat unclear to justify the usage of selective model for the evaluations of the uncertainty. The original work for selective model was using it when the task has the option of 'rejection' instead of performing 'prediction', but it is not clear why it is related to the measurement of uncertainty.
- There is no new method, no new metric to be proposed in this work. In other words, the contribution of this work may be purely empirical.
- There are a number of concerns on the experiments in terms of fairness: 1) it is unclear what Figure 2 means (see the first point in the weakness). 2) There are so many factors to determine the performance of a network on ImageNet that were not considered in this work such as optimization related parameters and practices, random initialization leading to different performance, possible differences in loss functions used and so on. However, a number of observations seem to assume that these are similar among all 484 trained models.
- It is very hard to grasp the main theme of this work as a conference paper. Many observations were lined up, but what are the main contributions? It is not really surprising to see ViT performing better than EfficientNet-V2.

Comments:
- Some of the important contents are in the Appendices. Unfortunately, in this manuscript, missing some of them in the main text seems to make this work to be very hard to understand. I strongly recommend to revise the manuscript to be accessible to more readers by writing key ideas and contents in the main body.
- There are too many contents in the main body and it is very hard to grasp all of them. It will be great if the authors can write down clear contributions of this work in the beginning and some key conclusions / take-home messages of this work as remarks.
- In selective model, the definition of coverage does not contain f while the symbol contains it (phi(f, g)). In addition, how did the authors implement g in experiments? It seems that g is important to define in details.
- It is clear how MC dropout measures the uncertainty, but it is not clear how selective model does. Please elaborate more on this.
- In Page 6, it says "While TS is usually beneficial, it could harm some models (see Figures 3 and 4)." How can we argue it since ECE is also measured empirically. In other words, the model could be ok while the measured ECE could contain some measurement errors.
- In Page 9, how can we conclude that "providing reliable uncertainty estimation"? In other words, how one can measure reliability of uncertainty estimation and how were they measured in this work. It is not easy to see the supporting data / experiments for this sentence.



**Summary Of The Paper:**

This work presents a number of empirical comparisons for the metrics on uncertainty estimation performance such as AUROC, ECE with 484 ImageNet classification models and has drawn some observations such as the superior properties of distillation over vanilla training, pre-training and adversarial learning, the superiority if ViT over other networks, and so on.

**Summary Of The Review:**

Some of the observations are interesting from extensive experiments, but there are a number of concerns on the metric to be used (e.g., selective model for uncertainty). There is no new method presented in this work, which makes this contribution somewhat weak for top conferences like ICLR (but could be an interesting workshop paper). The conclusions are not really surprising or are not easy to believe since there are other factors that were not considered in this work.

---

> ### Author Response · Authors · 2021-11-11
> **Selective model and novelty part1**
>
> Thanks for your constructive feedback,
>
> *"It is somewhat unclear to justify the usage of selective model for the evaluations of the uncertainty. The original work for selective model was using it when the task has the option of 'rejection' instead of performing 'prediction', but it is not clear why it is related to the measurement of uncertainty."*
>
> (1) Selective prediction is commonly viewed as an application of uncertainty estimation [1, 2, 3]. The "engine" of selective prediction is a confidence function (kappa) such as MC-dropout, namely, an uncertainty estimator. The selective predictor decides when to abstain based on the ranking provided by an uncertainty estimator.
> As a consequence, good selective prediction performance can only be achieved when the underlying uncertainty estimator kappa has good performance in terms of ranking quality. In an ideal ranking of instances, inaccurate predictions should be ranked as low as possible, which can be evaluated by selective performance, regardless of whether or not we intend to reject any predictions. Please refer to the next point for further information.
> (2) As discussed in the paper, the metrics to evaluate uncertainty estimation are task-specific. the selective performance is *very* relevant to tasks in which the user is able to reject instances (such as the task presented in paragraph 2 of the intro). In those scenarios, the user might only care about how the uncertainty estimation performance translates to better accuracy (at the expense of it abstaining from predicting for some instances).
> While you are correct, it is not the only way to evaluate uncertainty estimation, and it can sometimes be irrelevant (e.g. the model cannot reject). This is why we also evaluate the calibration performance (measured by ECE) in all of our results. Users must decide for themselves which is more important for a given task.
>
> [1] Bias-Reduced Uncertainty Estimation for Deep Neural Classifiers, Yonatan Geifman, Guy Uziel, Ran El-Yaniv
>
> [2] Uncertainty-Aware Training of Neural Networks for Selective Medical Image Segmentation
>
> [3] Disrupting Deep Uncertainty Estimation Without Harming Accuracy
>
> *"It is clear how MC dropout measures the uncertainty, but it is not clear how selective model does. Please elaborate more on this."*
>
> The engine of a selective model can be any function that estimates uncertainty in a way that ranks predictions that are more likely to be incorrect higher than those that are more likely to be correct. As an example of such a scoring function, MC dropout measures predictive entropy (or variance in regression) for instances that are less likely to be correct. Selective models are as good as their scoring functions, which means they let us evaluate and the underlying scoring functions for ranking.
>
> *"There is no new method, no new metric to be proposed in this work. In other words, the contribution of this work may be purely empirical."*
>
> In deep learning, some of the greatest papers are purely empirical and contain no new methods or metrics. Observations based on solid empirical research should be welcomed if they generate interesting research questions. We believe our paper falls into this category.  A very impressive example of such a paper is "Understanding deep learning (still) requires rethinking generalization" https://dl.acm.org/doi/abs/10.1145/3446776
>
> Then again, our paper does contain a novel component that we believe is important.
> In the context of C-OOD, we developed, implemented, and demonstrated a method for building an OOD dataset, which is tailored specifically to the model being tested and includes severity levels. The paper discusses the advantages of using our method over other alternatives.
>
> *"it is unclear what Figure 2 means (see the first point in the weakness)."*
>
> Thank you, we will clarify this point in the revision.
> This figure considers ranking performance for tasks allowing rejection, which means that the uncertainty estimation could be evaluated by how well it could decrease the risk when given the opportunity to reject predictions. It serves two purposes:
> (1) Demonstrate that when rejection is possible, the overall accuracy is not a good predictor of performance after rejection. In that example, the (green) ViT-SAM, which has a significantly lower overall accuracy (by 12%) than the (blue) EfficientNetV2 has nonetheless better accuracies when the possible rejection rate is 50\% of predictions or above that, and is the only model of the two that can provide an accuracy constraint of 95% or above for any coverage.
> (2) Explain the drawbacks of using AURC as a ranking metric instead of AUROC, or examining the entire RC curve.

---

> > ### Author Response · Authors · 2021-11-11
> > **Selective model and novelty part2**
> >
> > *"There are so many factors to determine the performance of a network on ImageNet that were not considered in this work such as optimization related parameters and practices, random initialization leading to different performance, possible differences in loss functions used and so on. However, a number of observations seem to assume that these are similar among all 484 trained models."*
> >
> > It is indeed true that all of these models were trained at least somewhat differently, and there are many factors to consider. Because we cannot train all of those models in a comparable way, we:
> > (1) Evaluated models that were comparable: such as ResNet50 models trained with various augmentations and loss functions and those without, distilled and undistilled models, models with pretraining or without, models trained with adversarial training or without etc. We listed some of the most significant factors, along with some less significant ones. We were surprised to find that most factors do not seem to have much impact on uncertainty estimation (such as different loss functions).
> > (2) The number of factors is incomprehensible and we cannot compare them all, so we looked for outliers and focused our comparison on them. ViT is one such example.
> >
> > *"It is very hard to grasp the main theme of this work as a conference paper. Many observations were lined up, but what are the main contributions?"*
> > *"There are too many contents in the main body and it is very hard to grasp all of them. It will be great if the authors can write down clear contributions of this work in the beginning and some key conclusions / take-home messages of this work as remarks."*
> >
> > Our contribution comprises a vast empirical study of the most significant factors affecting uncertainty estimation performance and a novel way to measure C-OOD performance. Those contributions are listed in the last paragraph of the intro, but thanks to your comment, we understand they need to be summarized and made clearer. Those contributions are as follows:
> > (1) The identification of knowledge distillation as an excellent training regime for uncertainty estimation. Bringing this fact to light could help future research on developing even better training regimes for uncertainty estimation.
> > (2) Identification of ViT as the naturally best architecture for uncertainty estimation (in both ID and C-OOD), regardless of its size, by a large margin. This fact could help future research to build better architectures for uncertainty estimation.
> > (3) Recognizing that TS also improves ranking (not just calibration). Future research on developing post-training strategies specialized in improving rankings may benefit from pointing out this fact.
> > (4) A novel method for constructing C-OOD datasets tailored to evaluate specific models is presented. Our intention is to publish our code for creating such datasets so future researchers can perform further research in this area (evaluating robustness, other ood detection methods we haven't considered, or improving performance). As we indicate in the paper (and visualize in Figure 6), the datasets and methods currently available for this task have limitations.
> >
> > *"It is not really surprising to see ViT performing better than EfficientNet-V2."*
> >
> > Why do you think EfficientNet-V2 should be worse than ViT in uncertainty estimation?
> > Despite this fact being less surprising for some, many other comparisons are highly surprising. It was surprising to see, for instance, that all transformers proposed as improvements to ViT (DeiT, CaiT, etc...) perform worse in terms of uncertainty estimation, regardless of their relative sizes. To us, the most surprising fact is that vision transformers (ViTs) are better than all convolutional neural networks in assessing uncertainty. Why should the inductive prior of transformers be better at this task?
> >
> > *"I strongly recommend to revise the manuscript to be accessible to more readers by writing key ideas and contents in the main body."*
> >
> > Thank you, we will improve this in the revision.
> >
> > *"In selective model, the definition of coverage does not contain f while the symbol contains it (phi(f, g)). In addition, how did the authors implement g in experiments? It seems that g is important to define in details."*
> >
> > Thank you, we will improve and fix those definitions. As for implementing selection function g, yes, we did. g rejects all predictions below a certain threshold, and you can define many thresholds for a single model. In the RC curves (Figure 2), for example, each point on the x axis is related to a different g (a selection function with a threshold resulting in an x coverage over the data). In our evaluations for a selective accuracy constraint (SAC) of 99% we used, we implemented the selection functions such that their thresholds would be the lowest possible, while still providing a 99% accuracy on the non-rejected predictions.

---

> > > ### Author Response · Authors · 2021-11-11
> > > **Selective model and novelty part3**
> > >
> > > *"In Page 6, it says "While TS is usually beneficial, it could harm some models (see Figures 3 and 4)." How can we argue it since ECE is also measured empirically. In other words, the model could be ok while the measured ECE could contain some measurement errors."*
> > >
> > > Firstly, it is important to clarify that the harm we mentioned is not a harm to the accuracy of the model, but rather to its calibration.
> > > Regarding the results. We calibrated the models on 5,000 instances of the validation, and then we tested them on 45,000 instances. These 45,000 test points should illustrate how well the model would behave in a real world deployment, and if these points were given at test time, the ECE measurements suggest that the model was less calibrated. Though there could be measurement errors, we have repeated the experiment for hundreds of models, and the effect was observed for about a hundred of these models. This makes measurement errors unlikely to be the cause.
> > >
> > > *"In Page 9, how can we conclude that "providing reliable uncertainty estimation"? In other words, how one can measure reliability of uncertainty estimation and how were they measured in this work. It is not easy to see the supporting data / experiments for this sentence."*
> > >
> > > Several metrics have been used to measure the reliability of uncertainty estimation:
> > > (1) ECE was used to measure probabilistic reliability.
> > > (2) AUROC was used to measure the reliability of the ranking (which means if instance A is correct and B is wrong, then what is the probability that confidence(A)>confidence(B)).
> > > (3) Furthermore, we have tested the outcome of using the uncertainty estimation by selective performance (for an accuracy constraint of 99%).
> > >
> > > All of these metrics are defined in Section 2. We further provided an example in the introduction of the importance of an uncertainty estimator good at ranking (which can be relied upon for selective performance in that case).

---

> > > > ### Comment · Reviewer_bFby · 2021-11-30
> > > > **I would like to thank the authors for their responses.**
> > > >
> > > > I would like to thank the authors for their responses. I appreciate that.
> > > >
> > > > I also agree with the authors about the value of the empirical papers such as "Understanding deep learning (still) requires rethinking generalization". However, while it has well controlled experiments to clearly show the proposed claims, this manuscript was not able to control all important parameters that can affect the performance of different methods. If all experiments are reasonably controlled, then this manuscript could be a wonderful contribution - however, in its current form, it is not easy to support its publication.
> > > >
> > > > Oh, if one can see uncertainty estimation as a task, then using most recent network such as ViT can be expected to outperform previous networks such as EfficientNet-V2. I think it is the authors' responsibility to contrast them - why is this result counter-intuitive?
> > > >
> > > > I think this manuscript has great potential, but its overall writing is very confusing with many missing descriptions and its experiments are not well-controlled to see the true effects of different metrics.

---

### Author Response · Authors · 2021-11-18
**Addressing our response**

Dear reviewers,

The discussion stage is nearing its end, but it hasn't been as active as we had hoped. We would appreciate it very much if you addressed our response.

Thanks,

The authors

---

### Decision · Program_Chairs · 2022-01-20

**Decision:**

Reject

**Comment:**

As an empirical paper, this paper studies uncertainty estimations with respect to various architectures and learning schemes. Three reviewers suggested acceptance based on the strength of the paper (fairly extensive experiments were conducted, and some new observations were discovered, such as the superiority of ViT). On the other hand, two reviewers proposed rejection due to lack of rigor in writing and lack of novelty. No consensus was reached through additional discussion. In particular, the reviewer's point that the experiment was not well controlled-different models were trained with different hyperparameters etc- seems quite important, and it weakens the significance of the contribution of the paper.

All reviewers agreed that it is a potentially interesting and important paper. I encourage the authors to resubmit in the future after carefully addressing the reviewers' concerns.